# Drosophila activins adapt gut size to food intake and promote regenerative growth

Christian F. Christensen[1], Quentin Laurichesse[1], Rihab Loudhaief[1], Julien Colombani ●[1,2] ✉ & Ditte S. Andersen ●[1,2] ✉

Rapidly renewable tissues adapt different strategies to cope with environmental insults. While tissue repair is associated with increased intestinal stem cell (ISC) proliferation and accelerated tissue turnover rates, reduced calorie intake triggers a homeostasis-breaking process causing adaptive resizing of the gut. Here we show that activins are key drivers of both adaptive and regenerative growth. Activin-β (Actβ) is produced by stem and progenitor cells in response to intestinal infections and stimulates ISC proliferation and turnover rates to promote tissue repair. Dawdle (Daw), a divergent Drosophila activin, signals through its receptor, Baboon, in progenitor cells to promote their maturation into enterocytes (ECs). Daw is dynamically regulated during starvation-refeeding cycles, where it couples nutrient intake with progenitor maturation and adaptive resizing of the gut. Our results highlight an activin-dependent mechanism coupling nutrient intake with progenitor-to-EC maturation to promote adaptive resizing of the gut and further establish activins as key regulators of adult tissue plasticity.

Adult tissues with high turnover rates depend on stem cells (SCs) to provide a continuous source of differentiated cells to maintain tissue homeostasis. Adaptation of organ size and physiology to internal or environmental cues involves adjustments of divisions and fate decisions to maintain tissue integrity and ensure organism health and reproductivity. Due to a high level of conservation of both structure, physiology, and the gene-regulatory network controlling ISC activity, the Drosophila midgut has proven a powerful model for dissecting the molecular mechanisms underlying adult SC activity and fate decisions[1,2]. Drosophila ISCs are embedded basally throughout the gut epithelium and divide non-symmetrically to give rise to a daughter ISCs and transient progenitor cells, enteroblasts (EBs) or EE progenitor cells (EEPs), that are destined to differentiate into ECs or EE cells (EECs), respectively[3–7]. In homeostatic conditions, the majority of ISCs undergo asymmetric divisions giving rise to one ISC (self-renewal) and one daughter cell committed to differentiate into an EB (90%) or pre-EEC (10%) ensuring a constant pool of progenitor cells[6,8]. By contrast, adaptive growth, which can be triggered by anticipatory internal signals, e.g., following mating[9–11], or environmental cues, is characterized

by accelerated ISC division rates and a shift from asymmetric to symmetric ISC divisions[12]. One striking example of adaptive growth is the shrinkage and expansion of the gut in response to cycles of starvation and refeeding, which has been observed in a wide range of organisms[12–18]. Widespread damage and severe intestinal infections also trigger a switch in division mode towards symmetric divisions to ensure rapid replacement of lost ECs[19–21]. To identify niche-derived signals controlling ISC division rates and/or fate decisions in conditions associated with accelerated tissue turnover rates, we used RNAis to knock down all secreted peptides (approx. 800) in different populations of the ISC niche and screened for reduced survival to enteric infection. Strikingly, among the top candidate hits, we identified two ligands belonging to the activin branch of the TGF-β superfamily, Actβ and Daw, and their inhibitor, Follistatin (FS), suggesting a key role of activin signaling in controlling ISC activity and fate decisions.

The architecture of the Activin signaling pathway is highly conserved between flies and mammals[22]. In mammals, activins and Nodal share the same receptors and effectors, and hence, the pathway is often referred to as the Nodal/Activin pathway. Binding of Nodal/

[1]Department of Biology, Faculty of Science, University of Copenhagen, Universitetsparken 15, Build. 3, 3rd floor, 2100 Copenhagen O Copenhagen, Denmark. [2]These authors contributed equally: Julien Colombani, Ditte S. Andersen. ✉e-mail: julien.colombani@bio.ku.dk; ditte.andersen@bio.ku.dk

Activin growth factors to two type II activin receptors (Activin receptor type IIA/B (ActRIIA/B)) results in the recruitment, phosphorylation, and activation of two type I activin receptors (Activin receptor-like kinase 4 and 7 (ALK4 or ALK7)), which subsequently triggers the phosphorylation of receptor-regulated Suppressor of Mother against Decapentaplegic (R-Smads) 2 and 3. Phosphorylated R-Smads associate with the Co-Smad, Smad 4, and translocate into the nucleus to regulate the expression of broad range of genes in a tissue-specific and context-dependent manner[23]. In much the same way, binding of Drosophila activins to a single type I receptor, Baboon (Babo), promotes the assembly of heteromeric type I/type II (Punt/Wishful thinking) receptor complexes triggering phosphorylation of the downstream activin-specific R-Smad-related protein, Smad on X (Smox), which together with the Co-Smad, Medea, translocates to the nucleus to regulate gene expression[22,24]. Alternative splicing of *babo* gives rise to three isoforms, Babo-A, Babo-B, and Babo-C, that differ in their extracellular domain, and hence their affinity for the three related activin ligands, Actβ, Daw, and Myoglianin (Myo)[24,25]. Daw, which is a divergent paralogue of Actβ more related to vertebrate TGFβ1, signals exclusively through Babo-C in vitro[25], whereas both Myo and Actβ was reported to utilize Babo-A[26–28]. For now, a clear relationship between Babo-B and one or more activins has not been established. Activin signaling is further regulated by the highly conserved secreted antagonist FS, which inhibits ligand/receptor interactions in both flies and mammals.

While the role of activin signaling in progenitor expansion and cell specification during early development and organogenesis is well established in both flies and mammals[23,24,29–39], its function in adult tissues is not well understood. For instance, Activin B/ActRIB signaling promotes stemness of hair follicle SCs in the skin[40], while neutralization of Activin A by the activin antagonist, FS, is required to maintain the proliferative potential of ΔNp63⁺ stem- and progenitor cells in mouse salivary glands[41]. In flies, FS is required in testicular somatic cyst SCs (CySCs) to maintain niche quiescence, as loss of FS in CySCs or ectopic activation of activin signaling in hub (niche) cells results in their transdifferentiation into CySCs and a loss of niche capacity[42]. It is not clear whether the opposing outcomes of activin signaling on SCs self-renewal and differentiation reflect tissue-specific requirements. Nevertheless, mutations affecting Activin/Nodal signaling are frequent in cancer stem cells (CSCs) in a variety of tissues[43–45], suggesting a broader role of activins in regulating SC fate decisions in adult tissues.

Here we show that activin signaling is instrumental in coordinating ISC proliferation and fate decisions in both homeostatic conditions and during regenerative/adaptive growth. While Actβ signals through Babo-A in ISCs to promote their self-renewal and differentiation into EBs, Daw/Babo-C signaling is required in EBs for their maturation into ECs. Actβ is produced by EBs in response to intestinal infections and required for tissue repair, while Daw expression is dynamically regulated by nutrient intake and plays a key role in the adaptive resizing of the gut during cycles of starvation and refeeding by controlling EB maturation and tissue turnover rates.

## Results

### Activin signaling regulates homeostatic gut epithelial turnover rates

While activins have essential functions during early development in mammals, their function in most adult tissue including the gut has not been studied. To investigate how activin signaling might regulate tissue turnover in the adult gut, we analyzed the expression pattern of the activin type I receptor, Babo, using a *babo* > UAS-GFP reporter and endogenously GFP-tagged Babo (Babo::GFP). We found that *babo* is broadly expressed, but enriched in ISCs and EBs (Fig. 1a–c and Supplementary Fig. 1a, a″). To analyze whether activin signaling is required to sustain tissue turnover in homeostatic conditions, we knocked down Babo and its downstream effector Smox in progenitor cells and

monitored the effect on gut size (Fig. 1d–g) and numbers of newly generated ECs using a Repressible Dual Differential stability cell Marker (ReDDM) line (Fig. 1h–l)[46]. The ReDDM system labels progenitor cells (ISCs/EBs) with long-lived Histone (His) 2B-RFP and a short-lived GFP, while newly generated ECs are only positive for His-2B-RFP that remains stable for 28 days after its expression is turned off in ECs (Fig. 1h). Knockout of either Babo or Smox reduced gut size (Fig. 1d–g) and knockdown slowed down the production of ECs, demonstrating that activin signaling is required in progenitor cells to maintain tissue turnover rates in homeostatic conditions (Fig. 1d–l). Consistent with this, ectopic expression of a constitutively active form of Smox in progenitor cells was sufficient to trigger ISC proliferation and accelerate tissue turnover (Fig. 1m–o′).

### Babo-A and Babo-C regulates different steps of ISC-to-EC differentiation

The *babo* gene encodes three different isoforms that differ in their extracellular domains and affinity for the three activin ligands. While Babo-C is thought to bind Daw, Babo-A was reported to mediate both Myo and Actβ signaling[25–28]. To analyze the role of each of these isoforms in regulating ISC-driven tissue turnover, we first analyzed their expression patterns. While isoforms A and C were expressed at intermediate and high levels, respectively, isoform B expression was hardly detectable in the gut (Fig. 2a). Knockdown of Babo-C reduced the number of ECs produced over time, showing that it is required for ISC proliferation and/or differentiation into ECs (Fig. 2b,d–e′, Supplementary Fig. 1b″). By contrast, Babo-A and Babo-B knockdown did not affect tissue turnover rates in homeostatic conditions, although Babo-A knockdown caused a significant reduction in the stem and progenitor pool (Fig. 2b, c, e, e′ and Supplementary Fig. 1b, b′). To analyze how Babo-A, Babo-B, and Babo-C depletion affects ISC differentiation we made use of a cell fate sensor (Cfs^ts) line allowing us to monitor the sizes of the ISC, EB and EC populations over time (Fig. 2f; ref. 47). Strikingly, knockdown of Babo-A resulted in a depletion of EBs, while the ISC pool remained stable (Fig. 2g–i″), suggesting that Babo-A regulates the maturation of ISCs into EBs. While Babo-B depletion did not affect the sizes of gut resident populations (Fig. 2g, g″), knockdown of Babo-C in ISCs/EBs caused an accumulation of both ISCs and EBs suggesting a defect in the maturation of EBs to ECs (Fig. 2g–h″, j, j″). Indeed, EB-specific Babo-C depletion triggered an accumulation of EBs, showing that Babo-C is required in EBs to promote their maturation into ECs (Fig. 2k, m, n). Consistent with Babo-C, and not Babo-A, signaling being rate limiting for tissue turnover rates in homeostatic conditions (Fig. 2b–e′), knockdown of all three iso-forms resulted in an accumulation of EBs (Supplementary Fig. 1c–e′). As expected, knockdown of Babo-A in ISCs, but not EBs, reduced EBs numbers (Fig. 2k–l, n–q). Hence, Babo-A and Babo-C display different expression patterns and control different steps of ISC to EC differentiation.

### Actβ/Babo-A signaling promotes regenerative growth

The role of activin signaling in regenerative growth has not been investigated in neither the mammalian or fly gut. We therefore tested how tissue repair triggered by oral infection is affected by ISC/EB-specific Babo knockdown. Knockdown of all three Babo isoforms significantly reduced the infection-induced ISC proliferative response (Fig. 3a, b) and led to a decrease in organism viability after oral infection of mated female flies with the pathogenic bacteria *Pseudomonas entomophila* (Pe; Fig. 3c). The reduced regenerative response was recapitulated by ISC/EB-specific knockdown of Babo-A, but not Babo-B or Babo-C (Fig. 3d, e). Using the cell fate sensor, we further showed that knockdown of Babo-A in progenitor cells specifically reduces the number of EBs, whereas ISC numbers are slightly increased compared with infected control guts (Fig. 3f–h′). Altogether, this is consistent with a key role of Babo-A in promoting ISC divisions and ISC-to-EB differentiation. Babo-A was previously reported to function

as a receptor for both Actβ and Myo. However, while Actβ expression was strongly upregulated in response to oral infection (Fig. 3i), the expression of Myo was only moderately increased (Supplementary Fig. 2a), suggesting that Actβ might be the principal driver of regenerative growth. We also observed an intermediate up-regulation of Daw 16 hours after infection (Supplementary Fig. 2a'). In homeostatic conditions, Actβ expression is detected exclusively in EECs (Fig. 3j, j"; ref. 27) but its expression is triggered in EBs in response to oral infections by the mildly pathogenic, *Erwinia carotovora carotovora* 15 (Ecc15) and the highly virulent Pe (Fig. 3k–m"). This is consistent with RNAseq data on FACS sorted gut cells showing a 100-fold upregulation of Actβ in EBs in response to oral *Pe* infection[48]. In accordance with EBs being the main source of Actβ fueling regenerative growth, we found that knockdown of Actβ in EBs, but not ISCs, EECs or ECs, significantly

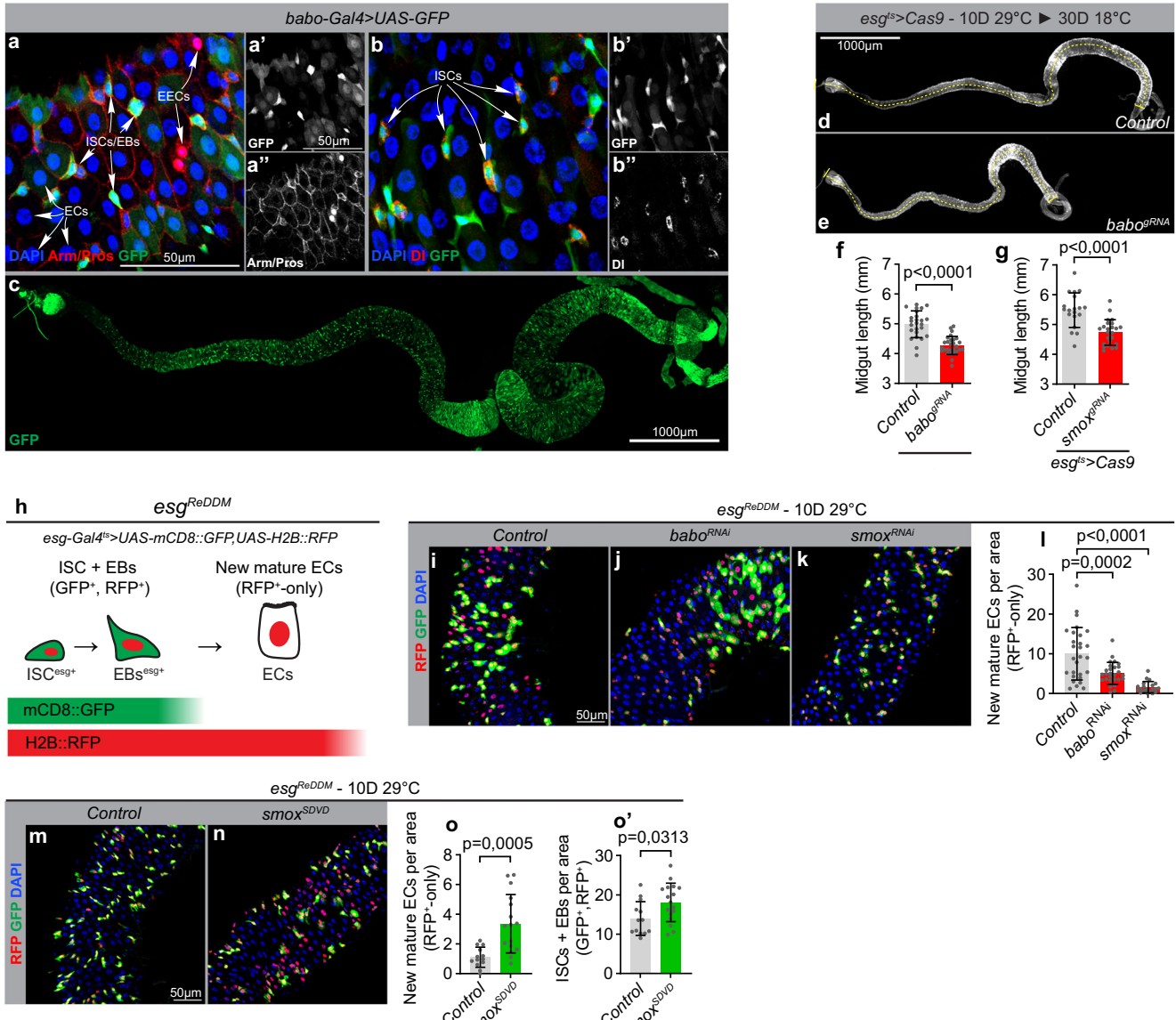

**Fig. 1 | The activin signaling pathway control turnover of intestinal cells.**
**a–c** Confocal image of a posterior (**a**, **b"**) or whole (**c**) midgut from a *babo*-Gal4>UAS-GFP reporter line stained for GFP (green, a-c), Armadillo (Arm) and Prospero (Pros) (Red, **a**, **a"**) or Dl (Red, **b**, **b"**), and DNA (blue, **a**, **b**) reveal enriched expression in delta-positive ISCs and other diploid cells (EBs) (**b**), not marked by nuclear Pros (**a**). This experiment was repeated independently 3 times with similar results. **d–g** Activin signaling maintains homeostatic gut size. Quantifications of midgut lengths after *escargot*-Gal4 (*esg>*)-mediated knockout of *babo* (**e–f**) (*n* = 25, 28 biologically independent guts) or *smox* (**g**) (*n* = 19, 24 biologically independent guts) in stem and progenitor cells using CRISPR/Cas9. **h** Schematic diagram of the *esg*[ReDDM] system. Membrane tethered CD8::GFP and nuclear localized H2B::RFP is specifically co-expressed in stem and progenitor cells with the *esg>* driver. As progenitors differentiate into mature cells, loss of *esg*-Gal4 expression terminates further production of GFP and RFP. Differential stability of the fluorophores results in rapid degradation of GFP whereas RFP remains in differentiated cells for an extended duration of time. For all ReDDM experiments, RFP-positive/GFP-negative polyploid cells were scored as newly produced ECs (**i–o'**) Activin signaling maintains homeostatic cell turnover. Representative confocal images of dissected posterior midguts from control flies (**i**, **m**) and flies expressing *babo*[RNAi] (**j**), *smox*[RNAi] (**k**), and *smox*[SDVD] (**n**) in stem and progenitor cells using *esg*[ReDDM] tracing for 10 days and quantification of cell turnover in each condition (**l**, **o**) (*n* = 28, 27, 20 and *n* = 13, 16 biologically independent guts). Expression of *babo*[RNAi] and *smox*[RNAi] reduces the number of new mature cells (RFP+-only) whereas expression a constitutively active *smox* construct (*smox*[SDVD]) increases the number relative to control. Expression of *smox*[SDVD] also expands the pool of stem and progenitor cells (GFP+, RFP+) relative to controls (**o'**) (*n* = 13, 17 biologically independent guts). Significance was tested with two-tailed unpaired t-tests (**f**, **g**, **o**, **o'**) and Kruskal Wallis (**l**) with post-hoc multiple comparison analysis. Data are presented as mean values ±SD. Source data are provided as a Source Data file.

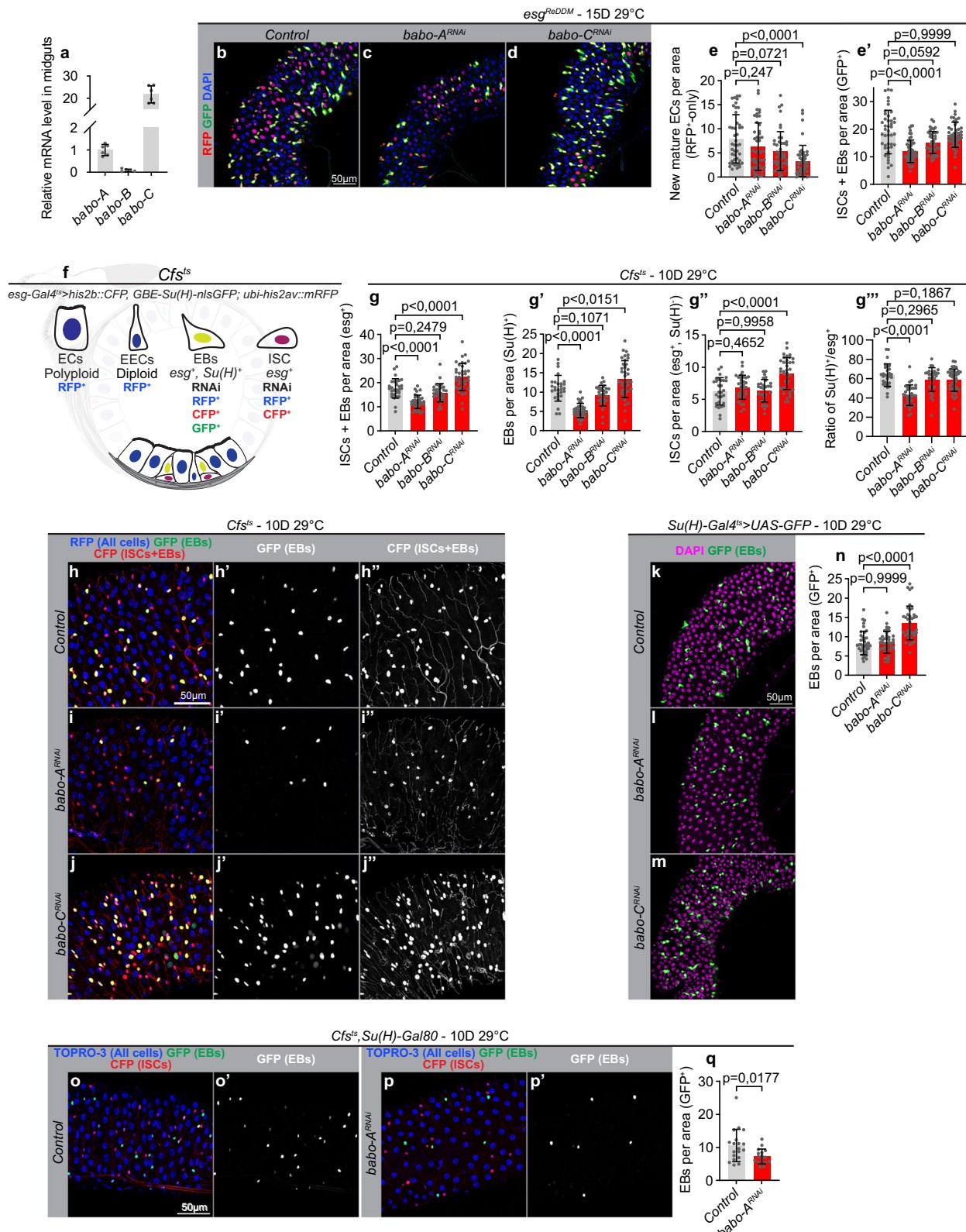

reduced the proliferative response (Fig. 3n, Supplementary Fig. 2b–e). Of note, knocking down Act-β in both ISC and EBs more efficiently suppressed the proliferative response to intestinal infections than EB-specific Act-β knockdown, suggesting that ISCs might constitute an additional source of Act-β (Fig. 3o–q). Consistent with this, Act-β was also upregulated in ISCs following Pe and Ecc15 infections (Fig. 3m′). Likewise, knockdown of Babo-A diminished the progenitor pool in

conditions of infection-induced regenerative growth and reduced tissue turnover rates (Fig. 3r, s, u, u′). Consistent with the role of Babo-C in promoting EB-to-EC maturation, knockdown of Babo-C increased the progenitor pool (Fig. 3u′) and suppressed EC replenishment in both homeostatic conditions and during tissue repair (Figs. 2e and 3r, t–u′). In agreement with a role of activin signaling in promoting tissue repair, ISC/EB-specific knockdown of Babo-A or EB-specific

**Fig. 2 | Babo isoforms A and C regulate different steps of ISC-to-EC differentiation. a** RT-qPCR analysis on dissected control midguts showing differential expression levels of the *babo-A*, *babo-B* and *babo-C* isoforms (*n* = 6, 5, 6). **b–e′** Quantification of tissue turnover in dissected posterior midguts from control flies and flies expressing *babo-A*^RNAi, *babo-B*^RNAi and *babo-C*^RNAi in stem and progenitor cells using *esg*^ReDDM tracing (pool of two experiments, *n* = 44, 47, 41, 45). **f** Schematic of the Cfs^ts system used to distinguish midgut cell populations. Stem and progenitor cells are collectively visualized by expressing nuclear localized UAS-his2b::CFP with *esg*-Gal4, EBs are marked by nuclear GBE-Su(H)-nlsGFP, while Ubi-his2av::mRFP causes all cells to express nuclear RFP. Thus, ISC and EBs are both marked by CFP, but only EBs are GFP-positive, whereas mature ECs and EECs are both marked by RFP but distinguished by nuclear size. **g–j″** Babo-A and Babo-C isoforms control distinct stages of ISC-EC differentiation. Representative confocal images of dissected posterior midguts from control flies (**h, h″**) and flies expressing *babo-A*^RNAi (**i, i″**) or *babo-C*^RNAi (**j, j″**) in stem and progenitor cells using Cfs^ts and quantification of cell type numbers in each condition (**g, g″′**) (pool of two experiments, *n* = 28, 33, 30, 38). **k–n** Babo-C is required in EBs to promote the EB-to-EC transition. Representative confocal images of dissected posterior midguts from control flies (**k**) and flies expressing *babo-A*^RNAi (**l**) and *babo-C*^RNAi (**m**) in EBs using Su(H)^ts > UAS-GFP and quantification of EB numbers in each condition (**n**) (pool of two experiments, *n* = 40, 40, 39). **o–q** Representative confocal images of dissected posterior midguts from control flies (**o, o′**) and flies expressing *babo-A*^RNAi (**p, p′**) in ISCs using Cfs^ts coupled to Su(H)-Gal80 and quantification of EBs (GFP-positive; q) (*n* = 19, 17). Significance was tested with a two-tailed Mann-Whitney test (**q**), one-way ANOVA (**g, g′, g″**) or Kruskal Wallis (**e, e′, g″′, n**) with post-hoc multiple comparison analysis. Data are presented as mean values ±SD. Source data are provided as a Source Data file.

knockdown of Actβ reduced the resistance of mated female flies to oxidative stress (Supplementary Fig. 2f–g).

Many signals that stimulate proliferation and tissue repair in young animals also contribute to aged-related dysplasia. To investigate how augmented levels of activin signaling might affect longevity, we knocked down the activin inhibitor, FS, which was reported to counteract activin signaling in both flies and mammals. We observed an increase in ISC proliferation upon EC-specific knockdown of FS in both young and old animals accompanied by a reduced lifespan of mated female flies (Supplementary Fig. 2h, i), suggesting that excess activin signaling might accelerate ageing-associated deterioration of the gut epithelium.

### Daw/Babo-C signaling adapts gut size to nutrient availability

The adult gut is a plastic organ that undergoes resizing in response to certain environmental cues. Hence, it was shown that the cycles of starvation and refeeding causes the gut to shrink and regrow, respectively, in a wide range of organisms[12–18]. Previous studies in flies have demonstrated an essential role of the insulin-like molecule Dilp3 in driving progenitor and gut expansion following the first meals in newly eclosed flies[12]. By contrast, it is not clear how short cycles of intermittent feeding affects progenitor number in the mature gut nor how adaptive growth is regulated in this condition. We therefore applied cycles of 48 hours of starvation followed by 24 h of refeeding to 6-day-old virgins. Two days of starvation was sufficient to shrink the gut, while 24 hours of refeeding allowed the gut to reach its pre-starvation size (Fig. 4a, b). Interestingly, while 2 days of starvation decreased the total number of ECs, EECs, and ISCs, it increased EB numbers (Fig. 4c, c″′). This contrasts with prolonged starvation, which results in a loss of EBs and massive apoptosis[12,17]. While we could detect moderate levels of apoptosis in ECs, we did not observe significant levels of apoptosis in ISC/EBs following 48 hours of starvation, suggesting that EBs are only lost upon prolonged starvation (Fig. 4d, d″, Supplementary Fig. 3a, b″). The accumulation of EBs observed after 48 hours of starvation recapitulated the phenotype caused by EB-specific knockdown of Babo-C (Fig. 2n), and hence, to test whether activin signaling regulates nutrient-dependent adaptive growth, we next monitored the expression levels of activin ligands during cycles of starvation and refeeding. Strikingly, we found that Daw expression was strongly suppressed by starvation and reexpressed upon refeeding (Fig. 4e and Supplementary Fig. 3c–f). Although the expressions of both Actβ and Myo were slightly reduced upon nutrient deprivation, their expression levels were not restored upon refeeding (Supplementary Fig. 3g, h). Consistent with the role of Daw in coupling adaptive resizing of the gut with nutrient availability, knockdown of its receptor, Babo-C, in progenitor cells reduced tissue turnover (Fig. 4j–l), prevented normalization of EB numbers (Fig. 4f–h), and impaired resizing of the gut in refed animals (Fig. 4i). This suggests that starvation-mediated repression of Daw signaling might shrink the gut by preventing EB differentiation, while refeeding promotes Daw-dependent EB maturation and adaptive growth. Consistent with this, Daw is required in ECs, where it is also highly expressed (Supplementary Fig. 3f), to resize the gut following refeeding by permitting the accumulated pool of EBs to mature into ECs (Fig. 4m–p). Intermittent cycles of starvation might be a frequently occurring phenomenon in nature, and hence, loss of gut plasticity could have far-reaching consequences for survival. To test this, we subjected flies to repetitive cycles of fasting and refeeding and monitored survival. Indeed, knockdown of Babo-C in progenitor cells or Daw in ECs is sufficient to make mated female flies more sensitive to successive cycles of starvation/refeeding (Fig. 4q, r), demonstrating the physiological importance of Daw/Babo-C signaling in coupling nutrient intake with adaptive gut resizing. Altogether, our work establishes activins as key regulators of adult gut homeostasis and identifies a role of activins in coupling tissue turnover rates and organ size with environmental cues to promote resistance to pathogenic infections and fluctuations in nutrient availability (Fig. 4s).

## Discussion

In mammals, the BMP branch of the TGF-β superfamily has a well-defined role in regulating intestinal homeostasis. BMP signaling exhibits a gradient along the crypt to villus axis. Low levels of BMP signaling at the bottom of the crypt is essential to promote Lgr5^+ ISC self-renewal, while high levels of BMP suppress proliferation and prevents expansion of the Lgr5^+ ISC population outside of the crypt[49,50]. The role of BMP/Dpp signaling in the fly gut is more complex, as different sources of Dpp was reported to promote[20,51] or restrict[52,53] regenerative growth. Interestingly, Ayyaz et al.[51] reported that Smox, normally considered an activin-specific mediator, was required downstream of hemocyte-derived Dpp to trigger regenerative growth. This was contradicted by a more recent study showing that hemocytes are dispensable for proliferative response triggered by intestinal infection, bringing into question whether Dpp mediates Smox-dependent regenerative growth[54]. According to the data presented here, an alternative possibility is that Smox-dependent regenerative growth is triggered by activins.

The intense research put into deciphering the role of BMP signaling in maintaining gut homeostasis contrasts with a lack of studies addressing the function of nodal/activin signaling in the adult gut. Here we show that activin signaling plays a key role in regulating epithelial turnover rates in the gut by controlling ISC proliferation and fate choices. While Actβ signals through Babo-A to promote ISC self-renewal and their differentiation into EBs, Daw/Babo-C signaling is required in EBs for their maturation into ECs. Intestinal infections induce high levels of Actβ expression in EBs, where Actβ is required for the accelerated epithelial turnover rates associated with regenerative growth. Daw expression is also induced by oral infections, and consistent with this, Daw/Babo-C signaling is required in EBs to replenish the pool of ECs in this condition. Interestingly, Actβ produced by the PNS acts on the hematopoietic progenitors in hematopoietic pockets

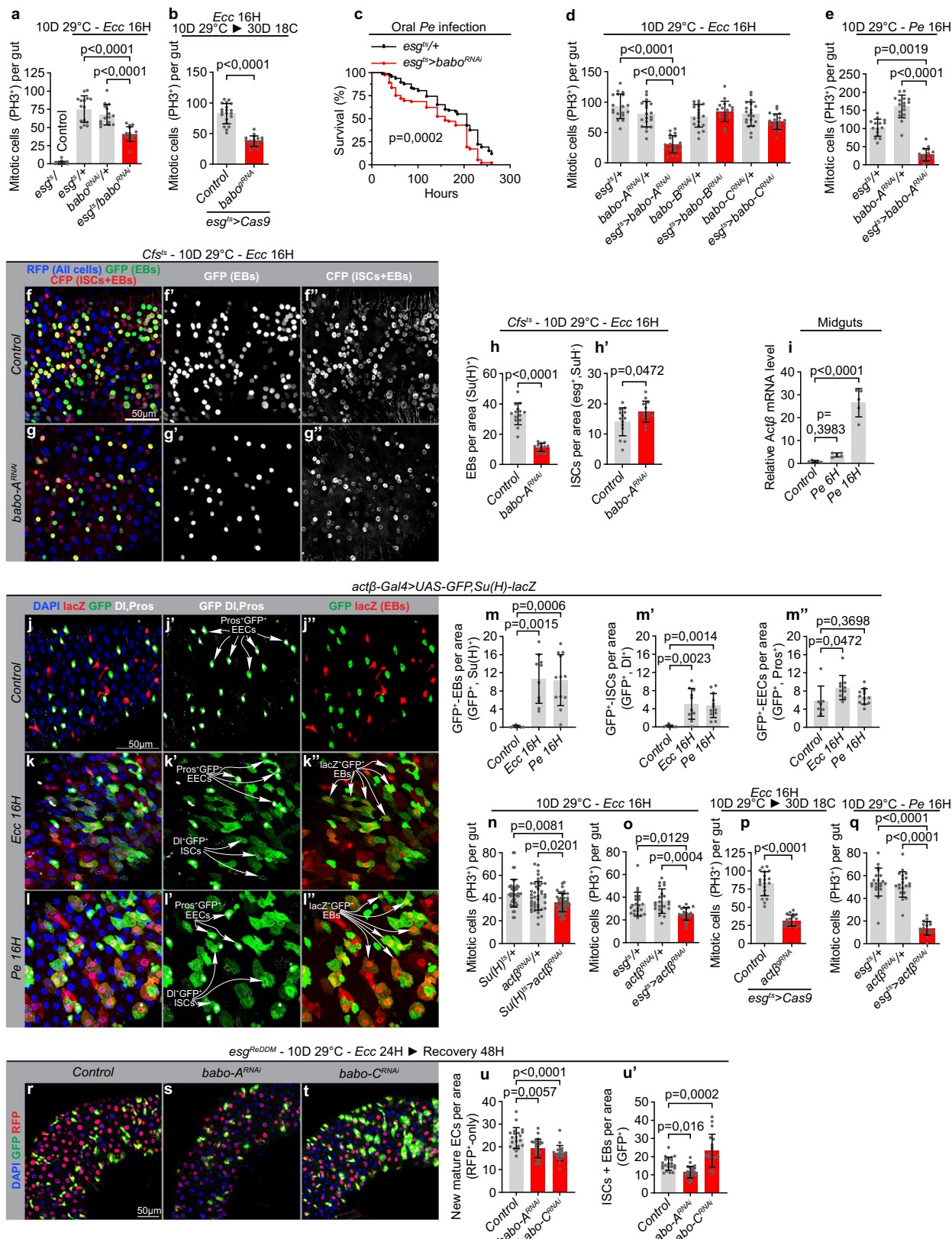

to stimulate their proliferation in larvae[55]. As the prime function of the PNS is to detect innocuous and noxious sensory stimuli, it is possible that activin signaling has a more general role in coupling harmful environmental cues with progenitor expansion to promote host defence. As mentioned before, Daw is related to vertebrate TGFβ1. In mammals, TGF-β/TβR1 signaling gradually increases from the crypt towards the tip of the villi with nuclear Smad3 being detectable in late TA and differentiated cells[56]. Based on in vitro and in vivo experiments, TGF-β is thought to promote terminal differentiation of ECs as these move out of the crypt and up the villi[57-61]. Moreover, TGF-ß signaling is required for tissue repair, as epithelial wide inactivation of TβRII results in the accumulation of undifferentiated progenitor cells[62]. Our

**Fig. 3 | EB-derived Actβ supports regenerative growth and tissue repair.**
**a**, **b** Quantification of PH3+ cells in midguts dissected from Ecc15 infected control flies and flies with ISC/EB-specific RNAi- (**a**; $n = 22, 15, 15, 15$) or CRISPR/Cas9-mediated (**b**; $n = 20, 18$) babo knockdown. **c** Mated flies with ISC/EB-specific knockdown of babo display reduced survival to oral Pe infection ($n = 98, 93$). **d**, **e** Quantification of PH3+ cells in midguts dissected from control flies and flies expressing babo-ARNAi, babo-BRNAi or babo-CRNAi in ISCs/EBs 16 h post Ecc15 (**d**; $n = 18, 20, 19, 20, 19, 20, 20$) or Pe (babo-ARNAi; **e**; $n = 15, 16, 17$) infection. **f**–**h′** Representative confocal images of dissected posterior midguts from control flies (**f**, **f″**) and flies expressing babo-ARNAi (**g**, **g″**) in ISCs/EBs 16 hours post Ecc15 infection, and quantification of cell populations (**h**, **h′** $n = 14, 13$). **i** RT-qPCR analysis on dissected midguts 6- and 16 hours post Pe infection ($n = 5, 5, 5$). **j**–**m″** Representative confocal images of dissected posterior midguts harboring actb-Gal4>UAS-GFP and Su(H)-lacZ (labeling EBs) stained for GFP (Green), lacZ (red),

Pros and Dl (white), and DNA (blue) 16 hours post Ecc15 (**k**, **k″**) or Pe (**l**, **l″**) infection, and quantification of EECs, ISCs, and EBs positive for GFP signal (**m**, **m″**; $n = 7, 10, 12$). **n**–**q** Quantification of PH3+ cells in midguts dissected from control flies and flies expressing actβRNAi in EBs (**n**; pool of two experiments, $n = 23, 45, 45$) or actβRNAi (**o**, **q**; $n = 22, 21, 22$ and $n = 20, 19, 20$) or Cas9 + actβgRNA (**p**; $n = 20, 19$) in ISC+EBs (esgts) 16 hours post Ecc15 (**n**–**p**) or Pe (**q**) infection. **r**–**u′** Representative confocal images of dissected posterior midguts from control flies (**r**) and flies expressing babo-ARNAi (**s**) and babo-CRNAi (**t**) in ISCs/EBs 48 hours (16H on Ecc15 + 32H recovery on normal food) post Ecc15 infection and quantification of cell turnover using esgReDDM tracing for 10 days (**u**, **u′**; $n = 21, 25, 19$). Significance was tested with two-tailed unpaired $t$-tests (**b**, **h**, **h′**,**p**), one-way ANOVA (**a**, **d**, **e**, **i**, **n**, **u′**) or Kruskal Wallis (**m**, **m′**, **m″**, **o**, **q**, **u**) with post-hoc multiple comparison analysis and a Mantel-cox Log-rank test (**c**). Data are presented as mean values ±SD. Source data are provided as a Source Data file.

data suggest that the role of Daw/TGFß signaling in promoting terminal EC differentiation might be conserved between flies and mammals.

The rapid tissue turnover observed in mammalian guts is an energy expensive process, consuming up to 15% of the total basal metabolic rate[63], and hence, during periods with reduced food intake, it is likely advantageous to reduce the total number of cells needing constant replenishment. Indeed, in both flies and mammals, prolonged periods of starvation can result in a reduction in gut size of up to 50%, which can be recovered following a few days of refeeding. In mammals, starvation slows down the proliferation in the transient amplifying region thereby reducing the number of epithelial cells, while +4 ISCs numbers and niche function is preserved[64,65]. This allows for rapid expansion of the epithelium upon refeeding. In flies, the expansion of progenitor cells in newly eclosed flies is triggered by food intake and mediated by the insulin-like molecule Dilp3[12]. By contrast, it is not clear how adaptive growth is triggered during cycles of starvation-refeeding in mature guts. So far, most starvation-refeeding studies have been performed within 3 days of eclosion, when progenitor cells have not reached homeostatic numbers[12,16], or involves prolonged periods of protein starvation of 7-15 days, which triggers high levels of apoptosis and dramatically reduces total numbers of EBs and ECs[17]. Importantly, as a proxy for intermittent feeding opportunities encountered by flies in the nature, it might be physiologically more relevant to study how shorter periods of starvation affects progenitor cell numbers. Here, we show that 24-48 hours of starvation, a condition that is not associated with massive cell death, results in an accumulation of EBs, despite a reduction in ECs and total number of cells. While the failure of EBs to differentiate into ECs results in shrinkage of the gut, it also generates a pool of poised EBs committed to promote adaptive growth upon refeeding. Indeed, EB numbers normalize shortly after refeeding, a process that depends on Daw/Babo-C signaling. Accordingly, Daw expression is strongly repressed upon nutrient deprivation, which likely restricts EB to EC differentiation, and reinduced upon refeeding. Altogether, this suggests that Daw/Babo-C signaling is essential for coupling adaptive resizing of the gut with nutrient intake. In mammals, nutrient restriction results in an accumulation of dormant ISCs (d-ISCs), also called +4 ISCs as they mark the transition from rapidly cycling Lgr5+ ISCs at the base of the crypt and the TA zone[64]. While d-ISCs cycle slowly or not at all in homeostatic conditions, they accumulate in response to nutrient restriction and participate in the regenerative growth during refeeding[64]. Hence, rapid regrowth of the gut upon calorie intake might rely on differentiation of stalled progenitors in both flies and mammals. The Daw-mediated adaptive resizing of the gut is physiologically relevant, as knockdown of Babo-C in EBs decreases the chances of survival of flies exposed to intermittent cycles of starvation and refeeding. Future research should aim at addressing whether TGF-β signaling plays a similar prominent role in adaptation of gut size to calorie intake in mammals.

While regulators of organ plasticity are essential for host adaptation, reproductivity and survival in an ever-changing environment,

the same signals are often deregulated in cancers. Indeed, mutations affecting TGF-β/Nodal/activin signaling has been observed in a wide range of cancers, and better understanding of how TGF-β/Nodal/activin signaling affects intestinal growth might provide a starting point for the development of therapeutic strategies targeting colorectal cancers.

## Methods

### Fly stocks and husbandry

Animals were maintained on a standard cornmeal diet (containing: 82 g/L cornmeal, 60 g/L sucrose, 34 g/L yeast, 8 g/L agar, 4.8 mL/L propionic acid and 1.6 g/L methyl-4- hydroxybenzoate) at 25 °C and 60% relative humidity under 12-hour light/dark cycle conditions. Virgin females were used for experiments unless otherwise stated. $w^{1118}$ was used as control for most experiments whereas the attP2 TRiP background stock was used as a control for TRiP RNAi lines. Temporal control of transgene expression using tubGal80^ts^ in adult flies was achieved by raising flies at 18 °C through development until 4–7 days after eclosion to allow maturation of the digestive system. Then, flies were shifted to 29 °C to induce Gal4 mediated transgene expression in a cell-type-specific manner. The duration of UAS induction was 10 days unless otherwise stated. Flies were flipped onto fresh medium every second day. For experiments using CRISPR/Cas9, UAS-Cas9.P2, and UAS-gRNA were co-induced at 29 °C for 10 days to allow mutagenesis of the target gene followed by a shift to 18 °C for 30 days.

The following lines were generous gifts from the colleagues in the fly community: Cell fate sensor, Cfs: w;*esg*-Gal4,UAS-CFP,Su(H)Gbe-nlsGFP; ubi-his2av-RFP (Lucy O'brien, Stanford University, USA). Myo1a-Gal4, Su(H)Gbe-Gal4 and Voila-Gal4 (Armel Gallet, ISA, France). *esg*-ReDDM UAS-CD8::GFP; UAS-H2B::RFP, tubGal80ts/TM6b (Maria Dominguez, IN, Spain). Su(H)-Gal80, TubGal80ts (Heinrich Jasper, Genentech, USA), UAS-dSmad2^SDVD^, Actβ-Gal4 and Daw-Gal4 (Michael O'Connor, CBS, USA). Actβ-gRNA v342095, *babo*-gRNA v341225, dSmad2-gRNA v342177, UAS-*babo*-RNAi v106092, UAS-Daw-RNAi v105309, Babo::GFP v318433 and UAS-Fs-RNAi v46260 were obtained from the Vienna Drosophila RNAi center. UAS-Actβ-RNAi BL29597, UAS-*babo*-RNAi BL29533, UAS-*babo*-A-RNAi BL44400, UAS-*babo*-B-RNAi BL44401, UAS-*babo*-C-RNAi BL44402, UAS-dSmad2-RNAi BL41670, UAS-Fs-RNAi BL57394, *babo*-Gal4 BL83164, *esg*-Gal4 BL93857, Su(H)Gbe-LacZ BL83352, *esg*-lacZ BL10359, UAS-Cas9.P2 BL58986, UAS-GFP BL39760, tub-Gal80ts BL7107 and BL7108, da-gal4 BL95282, *mex*-Gal4 BL91368, TRiP attP2 background BL36303 were obtained from the Bloomington stock center.

Detailed information about the genotypes behind each Figure is presented in Supplementary Table 1.

### Longevity

Flies raised at 18 °C through development were allowed to emerge over 48 h and transferred to fresh media for an additional 48 h at 25 °C to mate. Mated female flies were then sorted into vials containing

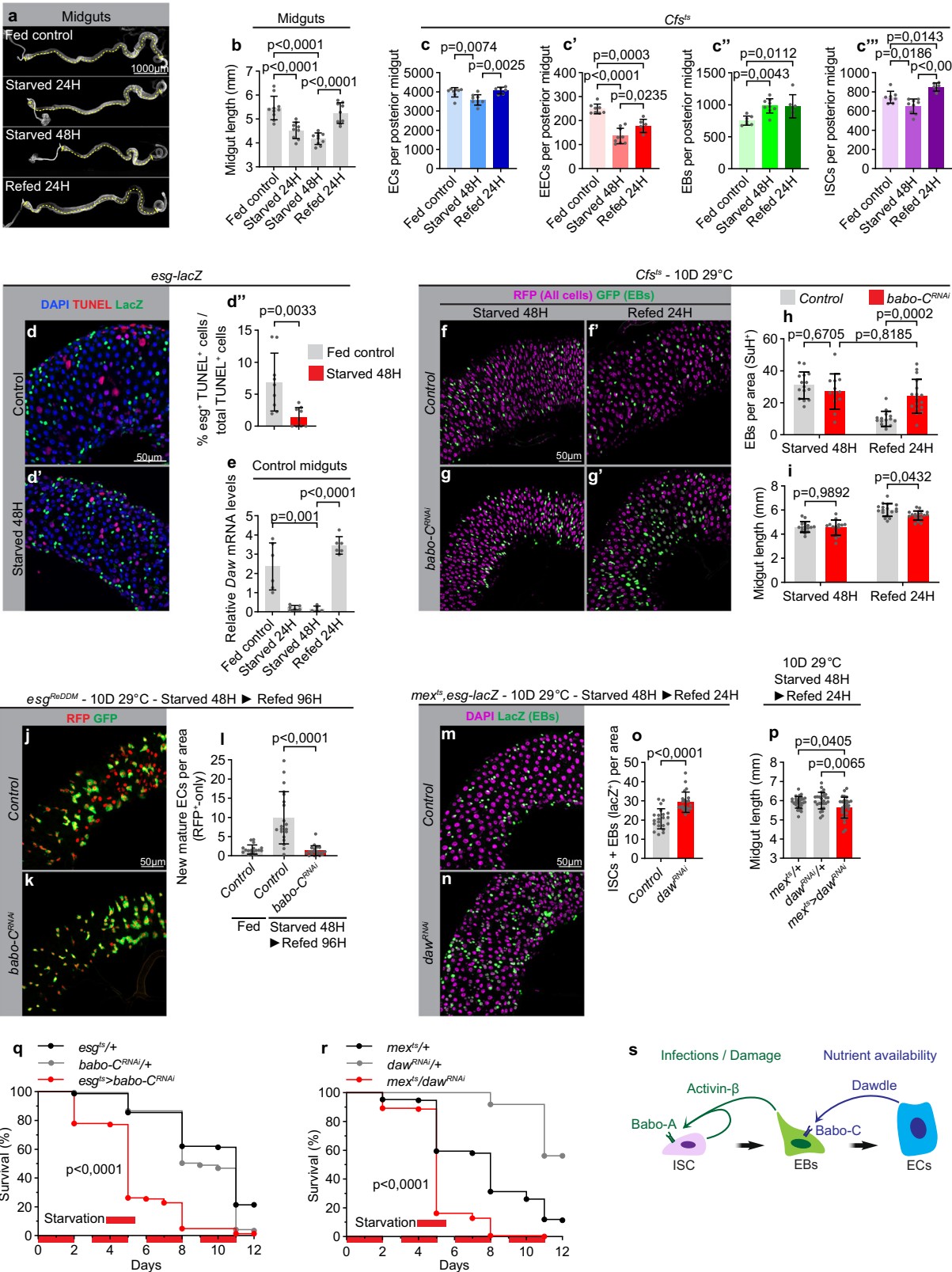

sugar/yeast food (consisting of 90 g/L sucrose, 80 g/L yeast, 10 g/L agar, 0,5% propionic acid, and 0.15% methyl-4-hydroxybenzoate) and transferred to 29 °C to induce transgenes. Flies were transferred onto fresh food every 2–3 days and the number of dead animals was assessed each 2–3 day until all animals had died. UAS-RNAi transgenes were backcrossed for at least 7 generations into the w[1118] background used as control.

## Oral infections

For oral infections with bacteria (*Ecc15* or *Pe*), overnight cultures were grown from single colonies (*Ecc15*) or directly from a glycerol stock kept at −80 °C (*Pe*). Cultures were grown in conical flasks containing LB broth at 29 °C (Ecc15) or at 30 °C with rifampicin supplied (*Pe*). The next day optical density OD600 was measured, cultures were spun down and the remaining bacterial pellet was

**Fig. 4 | Daw couples calorie intake with EB differentiation to promote adaptive growth. a, b** Representative images (**a**) and quantification of gut length (**b**; *n* = 10, 9, 10, 9) of dissected midguts of fed, 24H starved, 48H starved and 24H refed controls. **c, c′′′** Quantification of the total number of cells in the posterior midgut of fed, starved, and refed animals using Cfsts (*n* = 7, 9, 6). **d** Representative image of dissected midguts of control (**d**) 48H starved (**d′**) esg-lacZ animals labeled with TUNEL (in red) to visualize apoptotic cells. **d′′** Quantification of the proportion of TUNEL+ that are ISC/EBs (lacZ+) (*n* = 9,9). **e** RT-qPCR analysis on dissected control midguts in fed, 24H or 48H starved, and 48 H starved + refed flies (*n* = 6, 6, 5, 6). **f**–**i** Representative confocal images of dissected posterior midguts from control flies (**f, f′**) or flies with ISC/EB-specific knockdown of babo-C (**g, g′**) starved for 48 h (**f, g**) and refed for 24 hours (**f′, g′**). **h**–**i** Quantification of EB numbers (**h**) and midgut length (**i**) (*n* = 15, 15, 12, 17) in **f, g′**. **j**–**l** Quantification of cell turnover in dissected posterior midguts from fed, starved, or starved + 96H refed control flies (**j**) or flies with ISC/EB-specific knockdown of babo-CRNAi (**k**) using esgReDDM tracing for 10 days (**l**) (*n* = 18, 20, 20). **m**–**p** Quantification of ISC + EB numbers (**m**–**o**; *n* = 21, 23) and midgut lengths (**p**; *n* = 30, 30, 27) in dissected posterior midguts from 48H starved + 24H refed control flies (**m**) or flies with EC-specific daw knockdown (**n**). **q, r** Mated flies with ISC/EB-specific knockdown of Babo-C (**q**; *n* = 145, 147, 148) or EC-specific knockdown of daw (**r**; *n* = 150, 148, 149) show reduced resistance to successive cycles of starvation/refeeding. **s** Model of how activins regulate different steps of ISC-to-EC maturation in response to distinct environmental cues. Significance was tested with a two-tailed Mann–Whitney (**d′′, o**), one-way ANOVA (**b, c, c′, c′′, c′′′, e, l, p**) or two-way ANOVA (**h, i**) with post-hoc multiple comparison analysis and Mantel-cox Log-rank tests (**q, r**). Data are presented as mean values ±SD. Source data are provided as a Source Data file.

resuspended in an appropriate volume of 5% sucrose such that OD600 was adjusted to 200. The concentrated bacterial solution was then, in a volume of 50 µl, added directly onto Whatman filter paper discs resting on the surface of normal fly food with reduced yeast content (containing: 82 g/L cornmeal, 60 g/L sucrose, 17 g/L yeast, 8 g/L agar and 4.6 g/L methyl-4-hydroxybenzoate). To ensure efficient intake of bacteria, flies were sorted by 10 in empty vials and starved for 2 hours before transfer to the prepared vials containing bacteria. To assess survival to bacterial infection with *Pe*, mated female flies raised at 18 °C through development were sorted 10 flies per vial and transferred to 29 °C for 7 days before initiating the first infection. Flies were repeatedly infected once per day by transferring them to fresh vials with newly prepared *Pe* solution added. UAS-*babo*[RNAi] #106092 was backcrossed for at least 7 generations into the w[1118] background used as control.

### Starvation refeeding scheme

Flies were nutrient starved in vials containing $H_2O$ + 1% agar for 24 or 48 hours. For refeeding, 48 hours starved flies were transferred to normal food and allowed to feed for 24 hours. To assess survival to nutritional fluctuations, mated female flies raised at 18 °C through development were sorted 15 flies per vial and transferred to 29 °C for 4 days before initiating the first starvation-refeeding cycle. The number of dead flies were assessed once per day.

### Oxidative stress survival

To assess survival to oxidative stress, mated female flies raised at 18 °C through development were sorted 10 flies per vial and transferred to 29 °C for 15 days before oxidative stress challenge. Sugar/yeast food supplemented with 5% $H_2O_2$ was freshly prepared on the day. Flies were starved for 2 hours before being transferred to $H_2O_2$ food. The number of dead animals was assessed three times a day until all animals had died.

### Dissections and immunohistochemistry

Midguts were dissected in PBS and immediately transferred into fixative consisting of 4% paraformaldehyde in PBS for 1 hour at RT. Fixed midguts were washed twice with PBS and further two times 15 min with PBS + 0,1% Triton (PBS-T) with agitation. Midguts that were not stained with antibodies (*esg*[ReDDM] and Cfs[ts]) were immediately washed once with PBS and mounted on microscopy slides at this stage. Midguts that were stained with antibodies were incubated for 2 hours in blocking solution (PBS-T containing 10% FCS) followed by incubation with primary antibodies prepared in blocking solution at 4 °C overnight. The next day, midguts were washed three times 15 min with PBS-T and incubated with secondary antibodies for 3 hours at RT or overnight at 4 °C. At last, the stained midguts were washed two times 15 min with PBS-T followed by a single wash with PBS before being mounted on glass slides in Vectashield mounting media with or without DAPI. All midguts were mounted with 0,12 mm SecureSeal spacers (Grace Bio-Labs) except for experiments assessing the number of mitotic (PH3 + )

cells per gut. Primary antibodies were used with the following dilutions: rabbit anti-PH3 1:1000 (Milipore 06-570), chicken anti-GFP 1:10000 (Abcam 13970), mouse anti-β-Galactosidase (DSHB 40-1 A), mouse anti-Armadillo 1:100 (DSHB N2-7A1), mouse anti-Pros 1:200 (DSHB MR1A), mouse anti-Delta 1:100 (DSHB C594.9B) and rabbit anti-cDcp-1 1:100 (Cell Signaling Asp215). Secondary antibodies were used with the following dilutions: Alexa Fluor 488-conjugated goat anti-rabbit 1:1000 (Thermofisher #A-11008), Alexa Fluor 488-conjugated goat anti-chicken 1:1000 (Thermofisher #A-21467), Cy3-conjugated donkey anti-mouse 1:1000 (Jackson ImmunoResearch #715-165-151) and Cy5-conjugated donkey anti-chicken 1:1000 (Jackson ImmunoResearch #703-175-155). TOPRO-3 (Invitrogen) was used to counterstain nuclei when necessary.

### TUNEL assay

Apoptotic cells were detected using the TMR red In Situ Cell Detection Kit (Roche) according to the manufacturer's instructions. Midguts were dissected in PBS and transferred into fixative consisting of 4% paraformaldehyde in PBS for 20 min at RT. The fixed midguts were then handled as described for immunohistochemistry to counterstain with anti-GFP or anti-β-Galactosidase although with secondary antibodies diluted in the TUNEL reaction media and three extra PBS-T washes following the TUNEL reaction.

### Microscopy

Mounted midguts were imaged on an inverted Zeizz LSM-900 confocal microscope with Zen Blue software using 5x, 20x, or 40x objectives. For imaging CFP, GFP and RFP together (Cfs[ts]), the following bandwidth settings were used to avoid spectral bleed-through: CFP excitation at 405 nm and detection at 400–480 nm, GFP excitation at 488 nm and detection at 510-550 nm, RFP excitation at 561 nm and detection at 580-700 nm. 6-12 confocal stacks were acquired with an interval of 2 µm spanning the intestinal epithelium from the base to the lumen. An identical number of z-stacks were acquired for each experiment involving quantitative assessment of cell types. All images including the ones for quantifications were acquired in the posterior gut in region R4bc whereas the image in Fig. 1a is taken in region R5. Images were processed using Fiji and Adobe Photoshop software.

### Image analysis and quantifications

Image analysis and quantifications were performed using the open-source FIJI software. For quantitative measurements of specific cell types per area, a z-stacks were acquired with a 20x objective in the R4bc region for each mounted gut. Acquired z-stacks were converted to maximum-intensity projections and cell numbers was manually counted using the CellCounter Fiji plugin and normalized to the total epithelial area counted in. Cells per area is defined as cells per 10000 µm² in all cases. For quantitative measurements of total number of specific cell types in posterior midguts, tiles of z-stacks were acquired covering the entire posterior midgut from

top to halfway through the organ. Acquired z-stacks were converted to maximum-intensity projections and cell numbers was manually counted using the CellCounter Fiji plugin and multiplied by two. For quantifications of gut dimensions, tiled images of entire fixed midguts mounted with 0,12 mm spacers were acquired with fluorescence imaging with a 5x objective and stitched together in Zen Blue imaging software (Zeizz). Length was then measured in Fiji by drawing a segmented line using spline along the center of each midgut while R4 width was measured by drawing a straight line perpendicular to the midgut at the widest point in the R4 region. TUNEL$^+$ cells were determined in FIJI by applying the following steps to each maximum-intensity projection: (1) kuwahara filter= 5, (2) smooth, (3) auto threshold = max entropy, (4) convert to mask, (5) watershed, and (6) analyze particles.

## RNA extraction and qPCR

8-10 dissected midguts per biological replicate was directly transferred into lysis buffer and flash frozen in liquid nitrogen. Total RNA was extracted using RNAeasy microkit (Qiagen) according to the manufacturer's instructions. For cDNA synthesis, RNA was treated with DNase and reverse-transcribed using Superscript II reverse transcriptase (Invitrogen). The resulting cDNA was used for real-time RT-PCR on a QuantStudio 5 Real-Time PCR system using RealQ Plus 2x Master Mix Green (Ampliqon) with 8 ng of cDNA template. Samples were normalized to levels of ribosomal protein 49 expression levels and analyzed with QuantStudio 5 Real-Time PCR software using the delta-delta Ct method. Five to six biological replicates were used for each condition or genotype and triplicate measurements were performed.

The following primers were used:
actβ _F 5'-ACG GCA AAT TTT GAC AAA GC-3'
actβ_R 5'-TTG GTA TCA TTC GTC CAC CA-3'
daw _F 5'-TGA GCC ACC TCA TCC AAA TCA CCT-3'
daw_R 5'-TCG ATC ACG ATG AAT GGC CGG TAA-3'
Myo _F 5'-GGC GAC CAC ATA ATG ACT-3'
Myo_R 5'-TTA GCA TCA TCT CCC TGC ATT-3'
babo-A_F 5'-GGC TTT TGT TTC ACG TCC GTG GA-3'
babo-A_R 5'-CTG TTT CAA ATA TCC TCT TCA TAA TCT CAT-3'
babo-B_F 5'-GCA AGG ACA GGG ACT TCT G-3'
babo-B_R 5'-GGC ACA TAA TCT TGG ACG GAG-3'
babo-C_F 5'- GAC CAG TTG CCA CCT GAA GA-3'
babo-C_R 5'- TGG CAC ATA ATC TGG TAG GAC A-3'

## Statistics

Visualization of data in graphs and statistical analysis were performed using GraphPad Prism software. Each dataset was assessed for normal distribution using Shapiro-Wilks normality test before analysis. Comparisons of three or more groups were analyzed by one-way ANOVA or Kruskal–Wallis tests according to the outcome of normality tests followed by post-hoc multiple comparison analysis. Comparisons across genotype and condition were analyzed by two-way ANOVA followed by post-hoc multiple comparison analysis. Differences between control and one group was analyzed by unpaired t-tests if passing normality tests or Mann-Whitney tests if not. Comparisons of survival curves was analyzed using Mantel-Cox Log Rank test. All error bars indicate standard deviation.

## Reporting summary

Further information on research design is available in the Nature Portfolio Reporting Summary linked to this article.

## Data availability

All source data needed to evaluate the conclusions are present in the paper as a Source Data file. Source data are provided with this paper.

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

## Acknowledgements

We are grateful to all members of the Colombani and Andersen laboratory for scientific discussion and for carefully reading the manuscript. We thank M. O'Connor, M. Dominguez, A. Gallet, L. O'Brien, the Bloomington Stock Center and the Vienna Drosophila RNAi Center for fly stocks. J.C. and D.S.A. are funded by H2020 European Research Council grant number 803630, Novo Nordisk Foundation grant number

NNF180C0033920. We thank the Carlsberg foundation for an equipment grant CF19-0353.

## Author contributions

C.F.C., J.C., and D.S.A designed the research, C.F.C., Q.L, R.L, J.C., and D.S.A performed the genetic screen, C.F.C. conducted all the remaining experiments for the manuscript, C.F.C., J.C., and D.S.A analyzed the data, J.C. and D.S.A supervised the project, and D.S.A. wrote the manuscript.

## Competing interests

The authors declare no competing interests.
