## [Peer Review File · Nature Communications]

Drosophila activins adapt gut size to food intake and promote regenerative growthREVIEWER COMMENTS

Reviewer #1 (Remarks to the Author):

In their manuscript 'Drosophila activins adapt gut size to food intake and promote regenerative growth', Christensen et al. identified and report Act β as a key driver in adaptive and regenerative growth of the adult Drosophila midgut in a candidate gene screening approach. Act β signals from progenitors to drive ISC proliferation. Furthermore, they show that the activin Dawdle signals through Baboon receptors and Dawdle is regulated by food availability in turn adapting organ size to nutrient intake. The manuscript is well written and data is comprehensively presented. The data is based on state-of-the-art methods and provides new important insights in the mechanisms of organ size control and tissue plasticity. Although the authors present data on their findings, I have a few suggestions and concerns that I think should be approached experimentally and discussed in more detail in a revised version of this manuscript.

Major

1. *esg* and *babo* are recessive genes. In Figure and text, all gene names should be corrected upon Flybase nomenclature to avoid misinterpretation classics like *DI* (Delta, dominant) and *dl* (dorsal, recessive).
2. The proper function of RNAi stocks targeting *babo-A*, (-B) and -C is vital for this study as these lines are used throughout the study. Functional evidence of specific knockdown efficiency should be provided or references given in which these stocks were shown to provide functional knockdown.
3. The Cfsts method introduced in Fig2 is a versatile tool to address the main cell populations, but it is, as far as I understand it, too simplified how the authors display the method in Fig2C and whether it is actually giving a snapshot or tracing like information as the ReDDM method in Fig1. In the legend of Fig.2 the authors state that 'Stem and progenitor cells are collectively visualized by expressing nuclear localized UAS-*his2b::CFP* with *Escargot-Gal4*'. As *Cfs* also contains a *Gal80ts* repressor, new EEC and EC should be double positive for *his2b::CFP* (traced) and *ubi-RFP* after some days due to renewal? Or is the protein stability of *H2b::CFP* in nuclei of differentiating progenitors too low to persist into EC? Availability of information whether *babo-a* and -c (Fig.2F contains some RFP/CFP polyploid cells) manipulation still allows new EC differentiation could be quantified and support data in Fig.2B.
4. the authors discuss and exclude EB apoptosis upon short starvation as a mechanism for their observations. Although they discuss previous data from e.g. O'Brien in 2011 on EB apoptosis, they could consider extending their data or at least their discussion including newer findings from Arthurton (2020) and Reiff (2019). Apoptosis in the midgut, especially of EB, seems a fast process and these papers suggest that i) activation of a caspase (*Dcp-1*, Fig.4D) does not necessarily result in EB death and that ii) apoptosis needs to be approached directly with TUNEL and/or cleaved caspase 3 staining as caspases have non-apoptotic functions in EB differentiation.
5. The authors state (line 214-) that they detected apoptosis of EC. Is this statement derived from Fig.4D? Or another dataset? It would be interesting to know whether these *Dcp-1+EC* are the ones that are *dawdle-Gal4+* (Fig.S1F) as it would be suggestive for a positive or negative link between EC apoptosis and *dawdle* expression as shown for EGF signalling in Liang 2017.
6. As the authors introduce in the beginning, adaptive organ growth can be triggered by mating of

females flies. It was previously shown by the Dominguez, Aliaga, Reiff and Edgar labs (Reiff 2015, Ahmed 2020, Zipper 2020), that homeostatic renewal is significantly different in virgin females. In addition, virgin female food uptake is reduced (Cognigni 2011). In MM the authors state in the first paragraph that virgin females have been used, if not indicated differently. Given the role of mating, it is important to at least point out in text and fig legends of longevity, oral infections, starvation, esgreddm and ox stress assays whether virgins or mated females were investigated, which would also affect interpretation of experiments like midgut length/diameter (Fig.4B).

Minor

7. age of flies when dissected should be indicated in legends (e.g.Fig.2C...) and homogenous as ISC and EB behaviour changes in ageing flies.
8. X-axis descriptors switch between italic and normal in Figures.
9. the abbrev ISC is not introduced in the abstract
10. the authors switch between abbreviating prospero with pros and prosp.
11. fusion proteins are indicated differently (CD8::GFP and CD8-GFP)
12. Fig.3 M 'recovery' is spelled wrong
13. Fig.1C suggests that EB are the common precursor for EEC and EC. Although not relevant for the author's claims and the applied method, recent data from the Xi lab extends this view with a distinct precursor for EEC called EEP (Chen et al., 2018).

Reviewer #2 (Remarks to the Author):

In this manuscript from Christensen and colleagues, the authors investigate the mechanisms used by the *Drosophila melanogaster* midgut to adapt to changes in the environment such as infection and dietary changes. The authors first found that Activin signaling regulates homeostatic gut epithelial turnover rates. They further explore the effect of knocking down different isoforms of Babo, the activin receptor, on cell numbers in the gut, finding opposite roles of Babo-A and Babo-C on Enteroblast numbers. They then investigate the role of activin signaling in response to enteric infection, finding that babo-A in EB is required for infection-induced proliferation. Finally, the authors investigated the role of activin signaling in response to starvation-refeed cycles, finding that Activin signaling is again required for this process. While these findings are of interest and worthy of publication, they offer only an incremental increase in our understanding of the mechanisms regulating the *Drosophila* midgut. The response of the *drosophila* midgut to infection and diet has already been explored, as indicated in the papers cited by the authors themselves. Similarly, a role for activin signaling in the larval *drosophila* midgut has already been published (1999 *Genes and Dev* 10.1101/gad.13.1.98). The fact that Smox, the TF downstream of Activin signaling was needed in ISCs in the adult *drosophila* midgut was already known from Ayyaz 2015 *Nature cell Bio*. This publication is indeed cited, but somehow mentioned only in the discussion. Overall, while we find that these findings are mostly supported by the data (with some caveats, see comments below), we do not think there is enough novelty in these findings to support publication in *Nature communications*.

Major comments

There are a couple of miss-citations of finding from the literature. The authors should have a look and be sure of the data that they are citing.

For example:

55 – 57 “Intestinal infections also trigger a switch in division mode towards symmetric divisions to ensure rapid replacement of lost ECs (Hu and Jasper 2019; Tian, Wang, and Jiang 2017; Zhai, Boquete, and Lemaitre 2017).”

In the Hu 2019 publication, in response to Ecc15 oral infection the authors see more Asymmetric, not symmetric, outcome. There is an increased symmetric outcome only with Paraquat. In the Tian 2017 publication the symmetry-asymmetry assay is only performed upon feeding with Bleomycin. Pe is used for other assays. The Zhai 2017 publication is the only one suggesting a switch to symmetric division after infection. Maybe this complexity should be addressed here in the introduction (or in any case the sentence adjusted to the actual findings of the cited publications).

294 “So far, most starvation refeeding studies have been performed within 2 days of eclosion, when progenitor cells are still sparse (Lucchetta and Ohlstein 2017; O'Brien et al. 2011; Bonfini et al. 2021),” Lucchetta waited 3 days, Bonfini’s flies were kept on High Yeast for either 7,14 or 21 day before switching on High Sugar diet.

Regarding the methods:

469: “Virgin females were used for experiments unless otherwise stated.” Considering the importance of mating status for the size of the midgut and its activity, it would be nice if it was clearly stated (maybe somewhere in the figure legend) what experiment was done with Un-mated flies and what experiment was done with Mated flies. This will greatly improve the reproducibility of the manuscript.

470: “w1118 was used as control.”. Is there a reason for this choice? Considering the recent Nature metabolism from Sasaki (2021) which shows white regulating proliferative homeostasis in intestinal stem cells during aging, is there a risk this is not an appropriate control for some of the lines? Especially considering that the background of many lines used is not in w1118 (RNAi line from Bloomington for example). Repeating some of the key experiments with Gene switch drivers or with the Auxin system (same flies used and same temperature, just with or without drugs, so no background problems), or crossing to appropriate controls, would solidify the manuscript.

Replicates: the number of biological replicates is stated only for qPCR. I cannot see this information anywhere else, and some charts have relatively few dots (n numbers), which make me think that these experiments may have not been done in triplicate. It would be great if it could be stated somewhere (in M&M or in figure legends) the number of replicates done, and considering the reproducibility crisis currently present in science, if at least 3 biological replicate per experiment can be performed (I think only few experiments may have this problem, considering the number of dots on charts).

Image analysis: 573 “Cells per area is 574 defined as cells per 10000 μm^2 in all cases.” Was this fixed area always filled with cells and randomly selected?

General comment: It feels sometimes that there is a mix of arguments used to support a conclusion, but that some data are not shown (a bit like a Punnett square where some of the cells are not filled). For example: an experiment is being done with a certain RNAi, and the next mention this experiment but with the experiment itself done with a different RNAi (especially for Babo RNAi vs Babo-a RNAi vs Babo-c RNAi). Or for example in Figure 3, Pe and Ecc15 being used interchangeably at different timepoints for different readouts. Or in Figure 4D showing an example of Dcp-1 stain in starved condition, but not showing what the stain looks like in pre-starvation condition as a comparison.

Regarding Figure 1:

A: While using a Gal4 can give us a good idea of expression patterns (especially using a CRIMIC line), it may not recapitulate the expression of the gene itself. In this case, the authors report that “125 is expressed at high levels in ISCs and EBs (Fig. 1A-A’) and at variable levels in ECs (Fig. 1A), while its expression was not detected in EECs (Fig. 1A)”. However, in Dutta et al. (<https://doi.org/10.1016/j.celrep.2015.06.009>) they see transcripts for babo to be expressed in all cell types, with the highest expression in EEC. Confirmation of Babo expression pattern via antibody or other means would improve the reliability of this result (and/or discrepancies should be discussed in any case in the discussion).

For the ReDDm system, how long does the mcd8 persist? I can see in figure 1E that there are big polyploid cells that are clearly not progenitor cells that have both RFP and GFP. Were these cells scored? Or were they not scored because they had both markers? I cannot find a detailed description of the quantification criteria used. Were only ECs scored or any mature cell? The text description says “134 slowed down the production of ECs,” but on the figure it only mentions mature cells.

Regarding Figure 2:

For B, in the text it says “Mature EC”, in the chart mature cells. Which one is it?

152: “Strikingly, knockdown of Babo-A resulted in a depletion of EBs, while the ISC pool remained stable (Fig 2D-E, G-G’”), suggesting that Babo-A is required in ISCs for their maturation into EBs.”
Do we know if it is the case? Could it be that Babo-A is needed for retention of EB? Would need an EsgReDDM with babo-A RNAi to see if the effect is the same as Babo-RNAi. I see later an EsgReDDM, but babo-A RNAi is present only in Challenged conditions. Additionally, H, I, J check if there is an EB specific role of babo-A and C. The same should be done using DeltaTS or Esg-Gal4 Su(H) Gal80, to support the claim for a role of babo-A in ISCs.

Additionally, for most of the manuscripts an RNAi targeting all Babo isoforms is used. Would be interesting to see the effect of that line on cell numbers, especially considering that Babo-A RNAi and Babo-C RNAi seem to go in opposite directions for some of the cell numbers.

Regarding Figure 3:

ActB.Babo-A signaling promotes regenerative growth “and led to a decrease in organism viability after oral infection with pathogenic bacteria (Fig. 3C).”

Specifying that this bacteria is Pe (different than Ecc15 used for the rest of the figure) would be more precise. Is there a reason for the way these 2 bacteria have been using interchangeably?

“Using the cell fate sensor, we further showed that 170 knockdown of Babo-A in progenitor cells specifically reduces the number of EBs, whereas ISC 171 numbers are slightly increased compared with infected control guts”. Following a bit the comment to the previous figure, it would be good to see what happens in this condition to EB to EC transition, to validate the role of Babo-A in progenitors. Are these EB simply not produced? Or they just differentiate as soon as they become EBs? Without that I am not sure if we can support the conclusion that “Altogether, 172 this is consistent with a key role of Babo-A in promoting ISC divisions and ISC-to-EB 173 differentiation.”

For H, are the original data (not just the fold change) presented in the manuscript? This would help understand what are the levels of ActivinBeta mRNA and how they relate to Myo’s one.

“In homeostatic conditions, Act-Beta 178 expression is detected exclusively in EECs (Fig. 3I, K-K’; (Song et al. 2017), but its expression is 179 triggered in EBs in response to oral infection (Fig. 3I-K’).” In Figure 3J, most of the cells marked with su(h)lacZ seem to be already differentiated EC, going by the size of the nuclei (just compare to 3I to see the difference to what EBs look like normally). I am not sure if this image is enough to show upregulation of Activin Beta in EBs alone, although it is true that the data from Dutta et al. support this conclusion, even if in that case the timepoint was at 48 hours, which is a completely different phase of the midgut regeneration.

“In accordance with EBs being the main source of Act β fuelling regenerative growth,” Going by the Dutta paper data, much more Activin B is produced by EEs in term of quantity (Eb go from 0.037 to 3.6852 (99.6 fold), while EEC go from 3.966 to 41.643 (10.5 fold). So, I am not sure about this conclusion of the authors. However, the authors do find that KO of activinBeta in EEC does not lead to decrease in ISC proliferation, such as the one seen with EsgTS and Su(H). It is possible that the total amount of ActivinBeta produced is less important than the fold activation. Or it is possible that Pe and Ecc15 trigger different responses of Activin (the authors seem to use them interchangeably across this figure; it would be nice to see if the response is the same).

Regarding Figure 4:

“By contrast, it is not clear how short cycles of intermittent feeding 208 affects progenitor number in the mature gut nor how adaptive growth is regulated in this 209 condition.”

Lucchetta had 3 days old guts in her experiments, at the timepoint where starvation started. Is there any relevant literature that you could cite to determine what can be considered a mature vs immature gut? In Bonfini et al. 2021 elife, an RNA-seq following midguts from just eclosed to 5 days post eclosion on High sugar or High Yeast diet seems to suggest that 3 days of growth is not too dissimilar compared to 5 days, going by the PCA in Figure 5A of that publication. Size of the gut shown in SupFig4 of the same publication shows that 3 days old gut are not too dissimilar from 5 days old gut in term of length. O’Brien

in her 2011 Cell publication has data point only for 0,1 and then 4 days so it is hard to see where the cell number plateaued. I suspect this type of growth is also dependent on the diet, Genotype and conditions used in the laboratory. What is it like in the author's hands? And can they support the message that they are the first doing this type of short starvation-refeed experiment in the mature midgut?

For Figure 4D, it would be nice to have a control stain in non-Starved conditions (pre-starvation) of DCP-1 Stain (and possibly a quantification).

REVIEWER COMMENTS

Reviewer #1 (Remarks to the Author):

In their manuscript 'Drosophila activins adapt gut size to food intake and promote regenerative growth', Christensen et al. identified and report Act3 as a key driver in adaptive and regenerative growth of the adult Drosophila midgut in a candidate gene screening approach. Act3 signals from progenitors to drive ISC proliferation. Furthermore, they show that the activin Dawdle signals through Baboon receptors and Dawdle is regulated by food availability in turn adapting organ size to nutrient intake. The manuscript is well written and data is comprehensively presented. The data is based on state-of-the-art methods and provides new important insights in the mechanisms of organ size control and tissue plasticity. Although the authors present data on their findings, I have a few suggestions and concerns that I think should be approached experimentally and discussed in more detail in a revised version of this manuscript.

Major

1. *esg* and *babo* are recessive genes. In Figure and text, all gene names should be corrected upon Flybase nomenclature to avoid misinterpretation classics like *DI* (Delta, dominant) and *dl* (dorsal, recessive).

The suggested changes have now been incorporated in the revised text and figures

2. The proper function of RNAi stocks targeting *babo-A*, (-B) and -C is vital for this study as these lines are used throughout the study. Functional evidence of specific knockdown efficiency should be provided or references given in which these stocks were shown to provide functional knockdown.

We have now tested the efficiency/specificity of the RNAi lines used to target the individual isoforms (see figure S1b-b"). Our qPCR analyses show that all three isoform-specific RNAi lines efficiently and specially reduce expression of the targeted isoform. Although the *Babo-A*- and *Babo-C*-specific RNAi lines somewhat reduces *Babo-B* levels, our data show that knockdown of *Babo-B*, which efficiently reduces *Babo-B* levels, does not affect tissue turnover rates nor alters the sizes of gut resident cell populations.

3. The *Cfsts* method introduced in Fig2 is a versatile tool to address the main cell populations, but it is, as far as I understand it, too simplified how the authors display the method in Fig2C and whether it is actually giving a snapshot or tracing like information as the *ReDDM* method in Fig1. In the legend of Fig.2 the authors state that 'Stem and progenitor cells are collectively visualized by expressing nuclear localized *UAS-his2b::CFP* with *Escargot-Gal4*'. As *Cfs* also contains a *Gal80ts* repressor, new EEC and EC should be double positive for *his2b::CFP* (traced) and ubi-RFP after some days due to renewal? Or is the protein stability of *H2b::CFP* in nuclei of differentiating progenitors too low to persist into EC? Availability of information whether *babo-a* and -c (Fig.2F contains some RFP/CFP polyploid cells) manipulation still allows new EC differentiation could be quantified and support data in Fig.2B.

To analyze whether the *his2b::CFP* protein might persist in ECs, we quantified the number of *RFP⁺/CFP⁺ GFP⁻* polyploid cells in control guts. We found 2 *RFP⁺/CFP⁺ GFP⁻* polyploid cells out of 2019 *CFP⁺* cells ($\approx 0,1\%$) suggesting that this is a very rare occurrence and is unlikely to represent normal renewal. Furthermore, our experiments using *Cfs^{ts}* coupled to *Su(H)-Gal80* revealed that suppression of *Gal4* activity in EBs effectively eliminated CFP in the EBs (figure 2 o-p'). Together, these analyses showed that *his2b::CFP* is not

long-lived and does not persist in ECs. This suggest that there is a difference in the stability of CFP- and RFP-tagged His-2b.

4. the authors discuss and exclude EB apoptosis upon short starvation as a mechanism for their observations. **Although they discuss previous data from e.g. O'Brien in 2011 on EB apoptosis, they could consider extending their data or at least their discussion including newer findings from Arthurton (2020) and Reiff (2019).** Apoptosis in the midgut, especially of EB, seems a fast process and these papers suggest that i) activation of a caspase (Dcp-1, Fig.4D) does not necessarily result in EB death and that ii) apoptosis needs to be approached directly with TUNEL and/or cleaved caspase 3 staining as caspases have non-apoptotic functions in EB differentiation.

Findings from the Arthurton (2020) and Reiff (2019) publications show that activation of the **initiator** caspase, Dronc, can result in both apoptotic and non-apoptotic outcomes. However, antibodies detecting cleaved Caspase 3 and Dcp1 should normally both indicate activation of **effector** caspases (drICE (Cas3) and Dcp1), and hence, apoptosis. However, we have added new data to Figures 4d-d" and S3a-b" showing stainings of both Dcp1- and tunel-positive cells in control and starved guts. In our conditions, we find similar results with Dcp1 and tunel stainings. Our data show that in both control and starved guts, apoptotic (tunel-positive) cells correspond to ECs. Furthermore, the number of apoptotic cells only slightly increases after 48 hours of starvation in agreement with the reduction in ECs being triggered by preventing EB-to-EC maturation (Figure 4c").

5. The authors state (line 214-) that they detected apoptosis of EC. Is this statement derived from Fig.4D? Or another dataset? It would be interesting to know whether these Dcp-1+-EC are the ones that are dawdle-Gal4+ (Fig.S1F) as it would be suggestive for a positive or negative link between EC apoptosis and dawdle expression as shown for EGF signalling in Liang 2017.

We have now additional data displaying Daw expression (using a Daw>GFP reporter) in control, starved and refed conditions in Figure S3c-e, which in line with our qPCR data (Figure 4e) shows that Daw expression is completely suppressed in starved conditions. As Daw expression is completely suppressed in starved guts, we infer that it does not promote apoptosis (detected as sporadically dying ECs with Tunel/Dcp-1 staining) in this condition.

6. As the authors introduce in the beginning, adaptive organ growth can be triggered by mating of females flies. It was previously shown by the Dominguez, Aliaga, Reiff and Edgar labs (Reiff 2015, Ahmed 2020, Zipper 2020), that homeostatic renewal is significantly different in virgin females. In addition, virgin female food uptake is reduced (Cognigni 2011). In MM the authors state in the first paragraph that virgin females have been used, if not indicated differently. Given the role of mating, it is important **to at least point out in text and fig legends of longevity, oral infections, starvation, esgreddm and ox stress assays** whether virgins or mated females were investigated, which would also affect interpretation of experiments like midgut length/diameter (Fig.4B).

Mated flies were used only for survival and longevity assays (Figure 3c, 4q-r, S2f-g, and S2i), which has now been indicated in the text and figure legends.

Minor

7. age of flies when dissected should be indicated in legends (e.g.Fig.2C...) and homogenous as ISC and EB behaviour changes in ageing flies.

For all experiments, comparisons were made between genotypes of the same age. We have now specified the ages of the flies in the figures.

8. X-axis descriptors switch between italic and normal in Figures.

We have now corrected this

9. the abbrev ISC is not introduced in the abstract **We**

have now introduced the abbrev for ISC in the abstract

10. the authors switch between abbreviating prospero with pros and

prosp. **We have now consistently changed this to pros**

11. fusion proteins are indicated differently (CD8::GFP and CD8-GFP)

We have now corrected CD8-GFP and H2B-RFP to CD8::GFP and H2B::RFP

12. Fig.3 M 'recovery' is spelled wrong

This has now been corrected

13. Fig.1C suggests that EB are the common precursor for EEC and EC. Although not relevant for the author's claims and the applied method, recent data from the Xi lab extends this view with a distinct precursor for EEC called EEP (Chen et al., 2018).

We have now omitted the EECs to better illustrate the concept of lineage tracing in the ISC-to-EB-to-EC lineage (Figure 1h)

Reviewer #2 (Remarks to the Author):

In this manuscript from Christensen and colleagues, the authors investigate the mechanisms used by the *Drosophila melanogaster* midgut to adapt to changes in the environment such as infection and dietary changes. The authors first found that Activin signaling regulates homeostatic gut epithelial turnover rates. They further explore the effect of knocking down different isoforms of Babo, the activin receptor, on cell numbers in the gut, finding opposite roles of Babo-A and Babo-C on Enteroblast numbers. They then investigate the role of activin signaling in response to enteric infection, finding that babo-A in EB is required for infection-induced proliferation. Finally, the authors investigated the role of activin signaling in response to starvation-refeed cycles, finding that Activin signaling is again required for this process. While these findings are of interest and worthy of publication, they offer only an incremental increase in our understanding of the mechanisms regulating the *Drosophila* midgut. The response of the *drosophila* midgut to infection and diet has already been explored, as indicated in the papers cited by the authors themselves.

Activin signaling is highly conserved and is known to play a key role in progenitor expansion and cell specification during early development in both flies and mammals. While mutations affecting Activin/Nodal

signaling are frequent in cancer stem cells (CSCs) in a variety of tissues, its role in controlling adult tissue homeostasis remains poorly understood. The current study identifies two distinct mechanisms by which Activin-dependent signaling couples environmental cues with adaptive tissue-scale responses, and hence, provides a framework for understanding how activin signaling controls adult tissue homeostasis and disease in mammals. Furthermore, the study identifies novel mechanisms underpinning nutrient-dependent gut resizing, which does not rely on loss of progenitor cells, as observed in previous studies, but results in the build-up of progenitor cells that are poised to mature into ECs upon refeeding. While previous studies have identified the role of insulin signaling in driving the expansion of the gut in response to the first meal of newly eclosed flies, this is the first description of signaling (Daw/Babo-C) coupling nutrient availability with shrinkage/expansion of mature guts (guts that have reached homeostatic ISC/EB/EC numbers).

Similarly, a role for activin signaling in the larval drosophila midgut has already been published (1999 Genes and Dev 10.1101/gad.13.1.98).

The study referred to here, describes the genetic and biochemical characterization of Baboon (Babo) and its association with and activation of the Smad2 homologue, Smox. The authors show that *babo* mutant animals die as late third instar larvae or early during pupal development, and that guts from *babo* mutant embryos and third instar larvae exhibit a normal morphology. While these observations suggest that Babo is dispensable for gut development (based on the observation of gut morphology), they do not address the functional requirement for Babo signaling in the adult gut. Furthermore, these studies were performed on whole mutant animals, and hence, the temporal and spatial requirements for Babo was not addressed. Finally, as these studies were performed several years before the identification of ISCs in fly gut, there were no existing tools allowing the authors to test the effects that activin signaling might have on SC-driven tissue regeneration and adaptation.

The fact that Smox, the TF downstream of Activin signaling was needed in ISCs in the adult drosophila midgut was already known from Ayyaz 2015 Nature cell Bio. This publication is indeed cited, but somehow mentioned only in the discussion.

As discussed in the manuscript, the Ayyaz 2015 Nature cell Biol paper suggested that Smox acts downstream of hemocyte-derived Dpp to trigger regenerative growth. Another study subsequently showed that hemocytes are dispensable for infection-induced regenerative growth (Chakrabarti et al 2016). According to the data presented in our manuscript, it is likely that Smox-dependent regenerative growth is driven by activins. While we show that Smox is also required for homeostatic turnover rates (Figure 1g, 1k-l), our manuscript focuses on the specific roles of activin ligands in coupling environmental stressors with adaptive response. We provide a thorough functional characterization of the source and regulations of these ligands as well as the specific receptors and cell-types targeted.

Overall, while we find that these findings are mostly supported by the data (with some caveats, see comments below), we do not think there is enough novelty in these findings to support publication in Nature communications.

Major comments

There are a couple of miss-citations of finding from the literature. The authors should have a look and be sure of the data that they are citing.

For example:

“Intestinal infections also trigger a switch in division mode towards symmetric divisions to ensure rapid replacement of lost ECs (Hu and Jasper 2019; Tian, Wang, and Jiang 2017; Zhai, Boquete, and Lemaitre 2017).” In the Hu 2019 publication, in response to Ecc15 oral infection the authors see more Asymmetric, not symmetric, outcome. There is an increased symmetric outcome only with Paraquat. In the Tian 2017 publication the symmetry-asymmetry assay is only performed upon feeding with Bleomycin. Pe is used for other assays. The Zhai 2017 publication is the only one suggesting a switch to symmetric division after infection. Maybe this complexity should be addressed here in the introduction (or in any case the sentence adjusted to the actual findings of the cited publications).

It is correct that while Tian 2017 and Zhai 2017 observe a switch to symmetric ISC divisions following Pe and Ecc15 infections, Hu and Jasper 2019 do not observe a switch in response to Ecc15 infections. In Hu and Jasper 2019, the authors argue that exposure to Paraquat inflicts a more severe stress condition, and hence, triggers symmetric divisions to accelerate tissue turnover. We have altered the text to:

“**Widespread damage and severe** intestinal infections also trigger a switch in division mode towards symmetric divisions to ensure rapid replacement of lost ECs (Hu and Jasper 2019; Tian, Wang, and Jiang 2017; Zhai, Boquete, and Lemaitre 2017).”

“So far, most starvation refeeding studies have been performed within 2 days of eclosion, when progenitor cells are still sparse (Lucchetta and Ohlstein 2017; O'Brien et al. 2011; Bonfini et al. 2021),” Lucchetta waited 3 days, Bonfini’s flies were kept on High Yeast for either 7,14 or 21 day before switching on High Sugar diet.

We thank the reviewer for pointing this out and have now corrected this to read: “So far, most starvation-refeeding studies have been performed within 3 days of eclosion, when progenitor cells **have not reached homeostatic numbers** (Lucchetta and Ohlstein 2017; O'Brien et al. 2011), or involves prolonged periods of protein starvation of 7-15 days, which triggers high levels of apoptosis and dramatically reduces total numbers of EBs and ECs (McLeod et al. 2010)” **changing the text from 2 to 3 days and omitting the Bonfini et al 2021 reference.**

Regarding the methods:

“Virgin females were used for experiments unless otherwise stated.” Considering the importance of mating status for the size of the midgut and its activity, it would be nice if it was clearly stated (maybe somewhere in the figure legend) what experiment was done with Unmated flies and what experiment was done with Mated flies. This will greatly improve the reproducibility of the manuscript.

Mated females were only employed for survival assays (Figure 3c, 4q-r, S2f-g, and S2i), and this has now been specified in the respective figure legends.

“w1118 was used as control.”. Is there a reason for this choice? Considering the recent Nature metabolism from Sasaki (2021) which shows white regulating proliferative homeostasis in intestinal stem cells during aging, is there a risk this is not an appropriate control for some of the lines? Especially considering that the background of many lines used is not in w1118 (RNAi line from Bloomington for example). Repeating some of the key experiments with Gene switch drivers or with the Auxin system (same flies used and same temperature, just with or without drugs, so no background problems), **or crossing to appropriate controls**, would solidify the manuscript.

We have now repeated the experiments where w1118 was used as a control of Bloomington lines, using either a KK line with w1118 as a control (Figures 3a, 4m-p, r) or the same TRIP line, but with the TRIP attP2 background control (Figures 1i-l; 3n-o,q, S2b-c,f,h). In all cases, we obtained similar results to our initial data. Although significant, the effect of knocking down Act-j3 in EBs with Su(H)-G4 on the proliferative response was a bit weaker with the new TRIP control (Su(H)>; Fig 3n) and not as efficient as knocking down Act-j3 in both ISCs and EBs (*esg*>; Fig 3o-q). While ISC-specific knockdown of Act-j3 does not reduce the proliferative response (*dl*>; Fig S2d), we cannot exclude that ISCs and EBs both constitute a source of Act-j3. Consistent with this, we find that Act-j3 is induced in both ISCs and EBs (Fig. 3m'). We have added this interpretation to the main text.

Replicates: the number of biological replicates is stated only for qPCR. I cannot see this information anywhere else, and some charts have relatively few dots (n numbers), which make me think that these experiments may have not been done in **triplicate**. It would be great if it could be stated somewhere (in **M&M** or in figure legends) the number of replicates done, and considering the **reproducibility** crisis currently present in science, if at least 3 biological replicate per experiment can be performed (I think only few experiments may have this problem, considering the number of dots on charts).

We have now repeated some of the experiments to add more biological samples to our figures. For each experiment, several biological replicates were done, and for each genotype in each condition between 10 and 50 guts were used (except for control guts in figure 3m-m', where only 7 guts were used, as the result was clear-cut)" as indicated in the figure legends.

Image analysis: "Cells per area defined as cells per 10000 μm^2 in all cases." Was this fixed area always filled with cells and randomly selected?

Yes, the area was always filled with cells and always positioned in region 4b-c to permit direct comparison between guts and across genotypes.

General comment: It feels sometimes that there is a mix of arguments used to support a conclusion, but that some data are not shown (a bit like a Punnet square where some of the cells are not filled). For example: an experiment is being done with a certain RNAi, and the next mention this experiment but with the experiment itself done with a different RNAi (especially for Babo RNAi vs Babo-a RNAi vs Babo-c RNAi).

Or for example in Figure 3, Pe and Ecc15 being used interchangeably at different timepoints for different readouts.

For figure 3, we used 16 hours of infection for all experiments (as this is where the proliferative response peaks) except for Figures 3R-U', where the aim is to follow tissue turnover rates over a longer period (72 hs). We have now repeated key experiments done to address the role of Babo-A and Act-j3 in regenerative growth and find that knockdown of Babo-A in ISCs/EBs (*esg*>) and act-j3 in ISC/EBs (*esg*>) suppresses the proliferative response triggered by both ECC15 and Pe infections (Figures 3D-E,O,Q).

Or in Figure 4D showing an example of Dcp-1 stain in starved condition, but not showing what the stain looks like in pre-starvation condition as a comparison.

We have now added images to visualize Dcp-1 and tunel staining in pre-starved and starved guts (Figures 4d-d" and S3a-b"). Our results show that 48 hours of starvation does not trigger apoptosis in stem and progenitor cells (diploid cells) and only mildly increases apoptosis in ECs (polyploid cells).

Regarding Figure 1:

A: While using a Gal4 can give us a good idea of expression patterns (especially using a CRIMIC line), it may not recapitulate the expression of the gene itself. In this case, the authors report that “125 is expressed at high levels in ISCs and EBs (Fig. 1A-A’) and at variable levels in ECs (Fig. 1A), while its expression was not detected in EECs (Fig. 1A).”. However, in Dutta et al. (<https://doi.org/10.1016/j.celrep.2015.06.009>) they see transcripts for *babo* to be expressed in all cell types, with the highest expression in EEC. Confirmation of *Babo* expression pattern via antibody or other means would improve the reliability of this result (and/or discrepancies should be discussed in any case in the discussion).

To substantiate our results with the Crimic line, we have now added an image showing the distribution of *Babo::GFP* (protein trap). In line with Dutta et al, *Babo::GFP* is broadly expressed although it is enriched in stem and progenitor cells (Figure S1a-a’). We have now modified the text to align it with the results obtained with the Crimic and *Babo::GFP* tools: “To investigate how activin signalling might regulate tissue turnover in the adult gut, we analysed the expression pattern of the activin type I receptor, *Babo*, using a *babo*>UAS-GFP reporter and endogenously GFP-tagged *Babo* (*Babo::GFP*). We found that *babo* is broadly expressed, but enriched in ISCs and EBs”

For the ReDDM system, how long does the *mcd8* persist? I can see in figure 1E that there are big polyploid cells that are clearly not progenitor cells that have both RFP and GFP. Were these cells scored? Or were they not scored because they had both markers? I cannot find a detailed description of the quantification criteria used. Were only EC scored or any mature cell? The text description says “134 slowed down the production of ECs,”, but on the figure it only mentions mature cells.

We have now changed the text on the figures from mature cells to mature ECs to clarify this point. The transition from EBs to ECs is gradual and we often find EBs with rather large nuclei that are most likely at a very late stage of EB-to-EC transition. For analyses employing the ReDDM system, all RFP and GFP double positive cells were excluded, and only cells labelled with RFP alone were counted as newly produced ECs – in line with previous publications (Antonello et al 2015). We have clarified this in the Figure legend (1h).

Regarding Figure 2:

For B, in the text it says “Mature EC”, in the chart mature cells. Which one is it?

We have changed the text in the figure to “Mature ECs”.

“Strikingly, knockdown of *Babo-A* resulted in a depletion of EBs, while the ISC pool remained stable (Fig 2D-E, G-G’), suggesting that *Babo-A* is required in ISCs for their maturation into EBs.” Do we know if it is the case? Could it be that *Babo-A* is needed for retention of EB? Would need an *EsgReDDM* with *babo-A* RNAi to see if the effect is the same as *Babo*-RNAi. I see later an *EsgReDDM*, but *babo-A* RNAi is present only in Challenged conditions.

We have performed the *esgReDDM babo-A* RNAi experiment in homeostatic conditions (see Figure 2e). If *Babo-A* was needed for retention of EBs, we would expect to see more newly generated ECs. This is not the case. If anything, knockdown of *Babo-A* in ISCs reduces tissue turnover rates, although not significantly (Fig. 2e). This shows that while Act-b/*Babo-A* signaling regulates ISC-to-EB maturation in homeostatic conditions, it is not rate limiting for tissue turnover rates in this condition.

Additionally, H, I, J check if there is a EB specific role of babo-A and C. The same should be done using **DeltaTS or Esg-Gal4 Su(H) Gal80**, to support the claim for a role of babo-A in ISCs.

We have now performed the requested experiment with *esg-Gal4 Su(H) Gal80>babo-A RNAi* and confirmed that knockdown of Babo-A in ISCs reduces the number of EBs (Figure 2q).

Additionally, for most of the manuscripts an RNAi targeting all Babo isoforms is used. Would be interesting to see the effect of that line on cell numbers, especially considering that Babo-A RNAi and Babo-C RNAi seem to go in opposite directions for some of the cell numbers.

We have now used the CFS to evaluate the effect of KD of all Babo isoforms on cell numbers and find that the number of EBs are increased in this condition (Figure S1e'). This suggest that knockdown of Babo-A in ISCs is not rate limiting for ISC divisions and ISC-to-EB differentiation in homeostatic conditions. Consistent with this, knockdown of Babo-A in ISCs/EBs does not significantly reduce tissue turnover rates in homeostatic conditions (Figure 2e). Importantly, and consistent with the observed infection-mediated upregulation of Act-j β in EBs, Act-j β /Babo-A signaling is required for the accelerated tissue turnover associated with intestinal infections (Figure 3u). By contrast, Babo-C/Daw signaling is required to mediate EB-to-EC differentiation in homeostatic conditions (Figure 2j), and consistent with this, knockdown of all three isoforms results in an accumulation of EBs (Figure S1d-e). We have added a couple of lines to the relevant paragraph to discuss this.

Regarding Figure 3:

ActB.Babo-A signaling promotes regenerative growth “and led to a decrease in organism viability after oral infection with pathogenic bacteria (Fig. 3C).” Specifying that this bacteria is Pe (different than Ecc15 used for the rest of the figure) would be more precise. Is there a reason for the way these 2 bacteria have been using interchangeably?

We have now specified that we use Pe for survival assay. The reasoning for this is that ECC15 is a natural opportunist and only mildly pathogenic to Drosophila, and hence, Pe is often used in survival assays (reviewed in Buchon et al 2013). As mentioned above, knocking down Babo-A in ISCs/EBs and Act-j β in ISCs/EBs reduces the proliferative response to both ECC15 and Pe infections (Figure 3d-e).

“Using the cell fate sensor, we further showed that knockdown of Babo-A in progenitor cells specifically reduces the number of EBs, whereas ISC numbers are slightly increased compared with infected control guts”. Following a bit the comment to the previous figure, it would be good to see what happens in this condition to EB to EC transition, to validate the role of Babo-A in progenitors. Are these EB simply not produced? Or they just differentiate as soon as they become EBs? Without that I am not sure if we can support the conclusion that “Altogether, this is consistent with a key role of Babo-A in promoting ISC divisions and ISC-to-EB differentiation.”

As we find that the number of newly produced ECs is slightly (although not significantly) reduced upon ISC/EB-specific Babo-A knockdown in homeostatic conditions (Figure 2e) and reduced in regenerative conditions (using the ReDDM system, Figure 3u), we infer that the reduction in EBs is caused by fewer EBs being produced and not by accelerated EB-to-EC differentiation.

For H, are the original data (not just the fold change) presented in the manuscript? This would help understand what are the levels of ActivinBeta mRNA and how they relate to Myo's one. “In homeostatic conditions, Act-Beta expression is detected exclusively in EECs (Fig. 3I, K-K’;

(Song et al. 2017), but its expression is triggered in EBs in response to oral infection (Fig. 3I-K’).” In Figure 3J, most of the cells marked with su(h)lacZ seem to be already differentiated EC, going by the size of the nuclei (just compare to 3I to see the difference to what EBs look like normally). I am not sure if this image is enough to show upregulation of Activin Beta in EBs alone, although it is true that the data from Dutta et al. support this conclusion, even if in that case the timepoint was at 48 hours, which is a completely different phase of the midgut regeneration.

For Figures 3i and S2a-a’, only fold changes are indicated. We find that *Daw* is expressed that highest level followed by *Myo* and *Act-j3* in both homeostatic and regenerative conditions (see attached Figure 1). While *Daw* is most abundant in both conditions, it is not required for regenerative divisions.

Although homeostatic levels of *Myo* are higher than *Act-j3*, it is only mildly upregulated in response to *Pe* infection (3-fold), while *Act-j3* is strongly induced (30-fold) in response to *Pe* infection. Although we cannot exclude that *Myo* could also stimulate the regenerative response, knockdown of *Actj3* in stem and progenitor cells significantly reduces the proliferative response, and hence, *Act-j3* and *Myo* are not redundant.

As mentioned previously, these Su(H)-positive cells with large nuclei are usually scored as late EBs. Furthermore, our functional data support a role of *Act-j3* in ISCs/EBs, not ECs (Figure 3n-q and S2e).

“In accordance with EBs being the main source of $Act\beta$ fuelling regenerative growth,” Going by the Dutta paper data, much more Activin B is produced by EEs in term of quantity (Eb go from 0.037 to 3.6852 (99.6 fold), while EEC go from 3.966 to 41.643 (10.5 fold). So, I am not sure about this conclusion of the authors. However, the authors do find that KO of *activinBeta* in EEC does not lead to decrease in ISC proliferation, such as the one seen with *EsgTS* and *Su(H)*. It is possible that the total amount of *ActivinBeta* produced is less important than the fold activation. Or it is possible that *Pe* and *Ecc15* trigger different responses of *Activin* (the authors seem to use them interchangeably across this figure; it would be nice to see if the response is the same).

We have repeated the experiment and find that *ECC15* and *Pe* infections both upregulate *Act-j3* expression in EBs and ISCs (See figure 3m-m’). It is possible that the proximity of the source of *Act-j3* to ISCs plays a role – many ISCs are closely associated with EBs - and that ISCs also constitute a source of *Act-j3* (See figures 3n-q). EEC-derived *Act-j3* might have a more prominent role in mediating systemic effects (Song et al 2017). As knockdown of *Act-j3* in EECs does not compromise the proliferative response to intestinal infections, and even increase the proliferative response triggered by *Pe* infection (Figure S2b-c), we conclude that EBs constitute the relevant source of *Act-j3* for infection-induced proliferative responses.

Regarding Figure 4:

“By contrast, it is not clear how short cycles of intermittent feeding affects progenitor number in the mature gut nor how adaptive growth is regulated in this condition.”

Lucchetta had 3 days old guts in her experiments, at the timepoint where starvation started. Is there any relevant literature that you could cite to determine what can be considered a mature vs immature gut? In Bonfini et al. 2021 *elife*, an RNA-seq following midguts from just eclosed to 5 days post eclosion on High sugar or High Yeast diet seems to suggest that 3 days of growth is not too dissimilar compared to 5 days, going by the PCA in Figure 5A of that publication. Size of the gut shown in SupFig4 of the same publication shows that 3 days old gut are not too dissimilar from 5 days old gut in term of length. O’Brien in her 2011 *Cell* publication has data point only for 0,1

and then 4 days so it is hard to see where the cell number plateaued. I suspect this type of growth is also dependent on the diet, Genotype and conditions used in the laboratory. What is it like in the author's hands? And can they support the message that they are the first doing this type of short starvation-refeed experiment in the mature midgut?

Luchetta et al monitored the number of ISCs per gut at 0, 2, 3, and 5 days after eclosion and observed a significant increase in ISCs between day 3 and 5, suggesting that homeostatic numbers have not been attained at day 3 (Luchetta et al Figure S1D). They further show that ISC numbers remain stable from 5 to 18 days after eclosion (Luchetta et al Figure S1D). In our experiments, we leave the gut to mature for 4-5 days at 18C, shift them to 29C to initiate gene knockdown for another 6 days, and then starve flies for 48 hours. While Luchetta and Ohlstein observe a reduction in ISC and EBs (approx. 50%) following 48 hours starvation of 3-day old guts, we observed an increase in EB numbers and a mild reduction in ISCs after 48 hours of starvation. This shows that 3-day old guts do not respond to starvation in the same way as 5-10-day old guts. This could either be due to steady-state phase cell numbers not having been obtained or fewer lipid stores, and hence energy, to support ISC divisions/differentiation. Either way, our results reveal a gut resizing mechanism that differ considerably from that observed in 3-day old animals. Instead of reducing stem and progenitor numbers, starvation results in the accumulation of EBs, which are poised for producing ECs upon refeeding. Furthermore, our data identifies a critical role of Dawdle in coupling nutrient intake with EB-to-EC differentiation and gut resizing.

For Figure 4D, it would be nice to have a control stain in non-Starved conditions (pre-starvation) of DCP-1 Stain (and possibly a quantification).

We have now added Dcp-1 and tunel stainings of pre-starved and starved guts. We did not observe significant starvation-induced apoptosis in diploid *esg*⁺ cells (Figures 4d-d'' and S3a-b'').

REVIEWERS' COMMENTS

Reviewer #1 (Remarks to the Author):

In this revised version of the manuscript, the authors addressed all remarks made during the reviewing process. Finally, I congratulate them for this excellent work.